# 🎓 GraDA: Gradient-Guided Knowledge Distillation for Domain Adaptation

## Abstract

In this paper, we explore **how to enhance student network performance in knowledge distillation (KD) for domain adaptation (DA)**. We identify two key factors impacting student performance under domain shift: **(1) the capability of the teacher network** and **(2) the effectiveness of the knowledge distillation strategy**. For the first factor, we integrate a Vision Transformer (ViT) as the feature extractor and our proposed Category-level Aggregation (CA) module as the classifier to construct the ViT+CA teacher network. This architecture leverages ViT's ability to capture detailed representations of individual images. Additionally, the CA module employs the message-passing mechanism of a graph convolutional network to promote intra-class relations and mitigate domain shift by grouping samples with similar class information. For the second factor, we leverage pseudo labels generated by the ViT+CA teacher to guide the gradient updates of the student network's parameters, aligning the student's behavior with that of the teacher. To optimize for efficient inference and reduced computational cost, we use a convolutional neural network (CNN) for feature extraction and a multilayer perceptron (MLP) as the classifier to build the CNN+MLP student network. Extensive experiments on various DA datasets demonstrate that our method significantly surpasses state-of-the-art approaches. Our code will be available soon.

## 1 Introduction

Domain adaptation (DA) has attracted significant attention in recent research due to its potential to mitigate domain shift (Ben-David et al., 2010) between source and target domains, enabling the transfer of knowledge from labeled source data to unlabeled target data. Traditional DA methods primarily rely on convolutional neural networks (CNNs) (Kayhan & van Gemert, 2020) to learn domain-invariant representations. However, studies (Li et al., 2017; Naseer et al., 2021) indicate that CNN-based models are highly sensitive to domain shift. Recently, DA approaches based on Vision Transformers (ViTs) (Yang et al., 2023; Xu et al., 2022) have demonstrated superior performance over CNN-based methods (Xiao et al., 2023; Yu & Lin, 2023). While these approaches mark significant progress, deploying ViT-based models in real-world applications remains challenging, especially in scenarios demanding rapid inference, minimal storage, and lower computational costs, such as on resource-constrained devices. In contrast, compact CNN models like ResNet18 and ResNet34 (He et al., 2016) are often preferred for their efficiency. This raises an intuitive question: '*how can we collaboratively leverage the strengths of these two models within a unified framework?*' Specifically, '*can we utilize the strong representational capability of the ViT-based model during training while exploiting the computational efficiency of the CNN-based model during inference?*' This balance would meet the demands for high performance with low computational cost. Knowledge distillation (KD) offers a promising strategy to address this concern described as: **Teacher (ViT)** $\xrightarrow{Method}$ **Student (CNN)**. Herein, the knowledge acquired by the ViT-based teacher model is transferred to a compact CNN-based student model. We identify two critical factors that directly impact the performance of the student model: ① **the ability of the ViT-based teacher** and ② **the effectiveness of the teaching method**.

To satisfy ①, the teacher must perform effectively on labeled source data, demonstrating low training loss and robustness to domain shift. Following prior DA methods (Xu et al., 2022; Yang et al.,

2023), we employ a ViT model as the feature extractor, leveraging its strong representational capacity. However, these methods typically use a multilayer perceptron (MLP) as the classification head, which may have limited generalization due to its inability to capture relational information among neighboring samples. To address this limitation, we propose a Category-level Aggregation (CA) module, inspired by graph convolutional networks (GCNs) (Kipf & Welling, 2017), as the classification head to form the ViT+CA teacher network. The CA module enhances the teacher network's generalization by effectively capturing *intra-class relations*. Specifically, it enriches source features extracted by the ViT-based model through a message-passing mechanism guided by ground-truth labels. Similarly, the CA module improves intra-class information in the target domain based on pseudo labels generated from unlabeled target data. Additionally, it constructs a cross-domain knowledge graph, aligning unlabeled target samples with labeled source samples by *class-aware feature alignment*, where pseudo labels and source ground-truth labels share the same categories. By doing so, the teacher network not only captures structural representations within both domains but also reduces the discrepancy between them.

Regarding ②, employing ViT and CNN in a teacher-student paradigm, it introduces a cross-architecture challenge due to their distinct mechanisms. CNN-based models capture local image features through convolutional operations (Kayhan & van Gemert, 2020), whereas ViT-based models, via self-attention mechanisms, effectively learn global information (Dosovitskiy et al., 2021). Therefore, applying a feature-based KD approach (Heo et al., 2019; Chen et al., 2021) with the ViT-CNN pair requires additional transformation steps. While logit distillation (Hinton, 2015; Huang et al., 2022) may serve as an alternative, traditional logit-based KD approaches typically align the teacher and student networks by focusing on specific model weights corresponding to regions within the logit space. Consequently, the performance of the knowledge distillation process remains suboptimal. To address this problem, we propose a KD method named **Gra**dient-Guided Knowledge Distillation for **D**omain **A**daptation (**GraDA**). Drawing inspiration from (Wang et al., 2022), this approach emphasizes gradient knowledge distillation, where all weights of the student network are considered, and the teacher network guides the

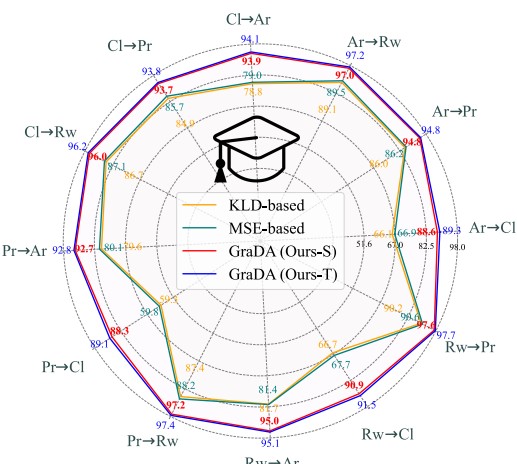

Figure 1: Comparison results of various knowledge distillation methods with our GraDA on **Office-Home** (Venkateswara et al., 2017) under UDA.

gradient direction to update the student's weights effectively. Specifically, in GraDA, the teacher network guides the student solely through pseudo labels, giving the student network flexibility to learn class representations on its own. This insight aligns with successful teaching strategies in education (Tan & Abbas, 2009), where teachers leave space for students to discover and solve problems on their own under guidance, rather than encouraging mechanical imitation. Moreover, teachers are expected to continuously expand their knowledge and teaching skills to provide higher-quality instruction. Notably, the student network in our method remains consistent with prior DA approaches (Jin et al., 2020; Li et al., 2021a), utilizing a CNN-based feature extractor and an MLP as the classifier (CNN+MLP). As illustrated in Fig. 1, our method is effective, particularly when the teacher and student networks yield similar classification results, surpassing existing logit-based methods.

In summary, our key contributions are three-fold:

- We design a strong teacher network that provides robust representations by enriching intra-class relations within each domain and mitigating domain shift across domains through class-aware feature alignment.

- We introduce gradient-guided knowledge distillation, allowing the student network to behave similarly to its teacher following its own capacity constraints, thus reducing cross-architecture and capability gaps.

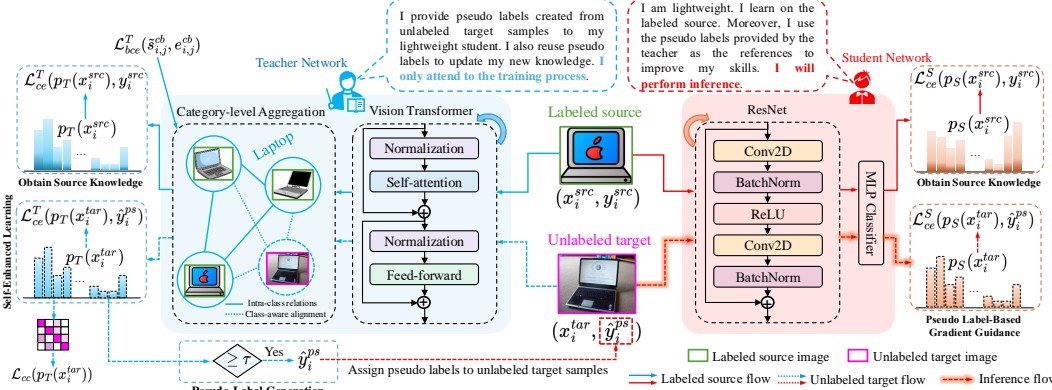

Figure 2: Illustration of the proposed GraDA. The teacher network includes a ViT-based model with a CA module, while the student network comprises a CNN-based model and an MLP. The teacher uses pseudo labels to guide the gradient direction for updating the student's parameters. Notably, the teacher is involved only during training, whereas the student is used for testing.

- Our proposed method is evaluated through quantitative and qualitative analyses, achieving state-of-the-art results across various DA tasks on popular datasets: **VisDA2017**, **Office-Home**, and **DomainNet**.

## 2 RELATED WORKS

**CNN/ViT-based in Domain Adaptation.** Traditional DA methods (Ganin et al., 2016; Saito et al., 2018) utilize convolutional neural networks (CNN) to learn domain-invariant and discriminative features. However, studies (Li et al., 2017; Naseer et al., 2021) have revealed that convolutional layers are sensitive to domain shift. More recently, ViT-based DA methods (Xu et al., 2022; Zhu et al., 2023a) have demonstrated that vision transformers (ViT) can effectively reduce the discrepancy between source and target domains, leading to significant improvements in performance. For instance, CDTrans (Xu et al., 2022) illustrates that the cross-attention mechanism within ViT can counteract domain shift. Consequently, ViT-based DA approaches can generate accurate pseudo labels that help mitigate domain shift via class-aware feature alignment. However, their memory-intensive attention mechanism makes them computationally costly and hard to deploy in real-world settings.

**Knowledge Distillation.** Over the past decade, KD methods have been mainly categorized into two types: logit-based (Hinton, 2015; Zhao et al., 2022; Huang et al., 2022) and feature-based (Heo et al., 2019; Romero et al., 2015; Chen et al., 2021). While logit-based approaches focus on narrowing the logit distribution between teacher and student networks, feature-based methods encourage the student to mimic the teacher's representations. However, these techniques, particularly feature-based approaches, struggle to transfer knowledge between networks with differing properties (Liu et al., 2022b), like ViT and CNN, due to low feature space similarity. (Zhu et al., 2023b) attempted to address this by using both feature- and logit-based mechanisms for cross-architecture knowledge distillation, though additional transformation steps are required. Nonetheless, both logit-based and feature-based methods operate in the same point-wise manner, where only specific parts of the student's weights are considered to match the teacher.

## 3 METHODOLOGY

### 3.1 PROBLEM FORMULATION

In unsupervised domain adaptation (UDA), we are given the source dataset $D_{src} = \{(x_i^{src}, y_i^{src})\}_{i=1}^{N_{src}}$, with $N_{src}$ representing the number of source samples. Each source image $x_i^{src}$ corresponds to an individual data point paired with a label $y_i^{src} \in [C]$. Here, $C \in \mathbb{Z}^+$ indicates the number of categories, and $[C]$ denotes the set $\{1, 2, \ldots, C\}$. Additionally, we are also provided with unlabeled target data, denoted as $D_{tar} = \{(x_i^{tar})\}_{i=1}^{N_{tar}}$, where $x_i^{tar}$ represents a target image,

and $N_{tar}$ denotes the number of target samples. The model is trained on both $D_{src}$ and $D_{tar}$, to achieve strong performance on $D_{tar}$. It is important to emphasize that $D_{src}$ and $D_{tar}$ share the same categories, and the target label $y_i^{tar} \in [C]$ is only used during the testing phase. The architecture and training process are illustrated in Fig. 2.

## 3.2 TEACHER NETWORK

**Architecture.** *To match* ①, we use ViT (Dosovitskiy et al., 2021) as the backbone ($f_{vit}$) for its superior global pattern capture via self-attention compared to CNNs (Kayhan & van Gemert, 2020). For the classification head, we introduce a Category-level Aggregation (CA) module, drawing inspiration from GCNs (Kipf & Welling, 2017), to enhance representations through message passing. The CA module comprises $f_{sim}$, which computes similarity scores, and $f_{agg}$, which aggregates feature vectors within a mini-batch.

**Operation.** During training, the input data is divided into multiple mini-batches of size $B$. Each training sample $x_i \in \mathbb{R}^{H \times W \times 3}$ is first encoded by $f_{vit}$: $z_i^{vit} = f_{vit}(x_i; \theta_{vit}) \in \mathbb{R}^d$, where $z_i^{vit}$ is a feature vector, $d$ is the embedding size, and $\theta_{vit}$ is the set of learnable parameters of $f_{vit}$. The batch of feature vectors $\{(z_i^{vit})\}_{i=1}^B$ is then processed by $f_{sim}$ and $f_{agg}$ in the CA module for feature aggregation. Specifically, $f_{sim} : \mathbb{R}^d \to \mathbb{R}^1$ is used to identify neighboring instances within the mini-batch by calculating similarity scores as:

$$\hat{s}_{i,j} = \texttt{sigmoid}\big(f_{sim}(\|z_i^{vit} - z_j^{vit}\|; \theta_{sim})\big), \tag{1}$$

where $\hat{s}_{i,j}$ is a scalar value that quantifies the level of relationship between the $i$-th and $j$-th feature vectors. $\theta_{sim}$ is the set of learned parameters of $f_{sim}$. The correlations among samples within a mini-batch are stored in the similarity matrix $\hat{S} \in \mathbb{R}^{B \times B}$, where $\hat{s}_{i,j} \in \hat{S}$. We normalize $\hat{S}$ by adding the self-connections formulated as follows:

$$\tilde{S} = D^{-\frac{1}{2}}(\hat{S} + I)D^{-\frac{1}{2}}, \tag{2}$$

where $I$ denotes the identity matrix, and $D$ represents the degree matrix of $\hat{S} + I$. Finally, the feature aggregation is processed as follows:

$$z_i^T = f_{agg}\Big([z_i^{vit}, \sum_{j \in B} \tilde{s}_{i,j} \cdot z_j^{vit}]; \theta_{agg}\Big), \tag{3}$$

where $z_i^T$ is an aggregated feature vector of the teacher network with the $C$-dimensional logit for the final prediction. $[\cdot]$ denotes the concatenation operation and $\tilde{s}_{i,j} \in \tilde{S}$. $f_{agg} : \mathbb{R}^{2d} \to \mathbb{R}^C$ is the linear projection and $\theta_{agg}$ is the set of learnable parameters of $f_{agg}$.

## 3.3 STUDENT NETWORK

**Architecture.** We attempt to build a straightforward network that meets the requirement for fast inference. Thus, we select the CNN-based model as the feature extractor, $f_{cnn}$, followed by an MLP as the classification head.

**Operation.** $f_{cnn}$ takes each mini-batch $\{(x_i)\}_{i=1}^B$ as input, and $x_i$ is encoded as $z_i^{cnn} = f_{cnn}(x_i; \theta_{cnn}) \in \mathbb{R}^{d'}$, where $z_i^{cnn}$ denotes the feature vector extracted by $f_{cnn}$ with the dimensional embedding of size $d'$, which is parameterized by $\theta_{cnn}$. Next, the MLP classifier processes $z_i^{cnn}$ to produce the predicted vector $p_S(x_i) = \texttt{softmax}(\text{MLP}(z_i^{cnn}; \theta_{mlp}))$, where $\theta_{mlp}$ is the set of learned parameters of MLP.

## 3.4 TRAINING STRATEGY FOR TEACHER NETWORK

We conduct a three-step approach in the teacher network: *1) Enriching Intra-Class Relations*, *2) Pseudo-Label Generation*, and *3) Self-Enhanced Learning*, intending to improve feature representations and mitigate the domain shift issue.

**Enriching Intra-Class Relation.** The teacher network exploits the relationships among labeled samples within each mini-batch $\{(x_i, y_i)\}_{i=1}^B$, thereby enhancing intra-class information. To be

specific, we train $f_{sim}$ to explore the pairwise similarity between the samples within the mini-batch using a binary cross-entropy (*bce*) loss as follows:

$$\mathcal{L}_{bce}^T(\tilde{s}_{i,j}, e_{i,j}) = -e_{i,j}\log(\tilde{s}_{i,j}) - (1 - e_{i,j})\log(1 - \tilde{s}_{i,j}), \tag{4}$$

where $e_{i,j}$ represents the ground-truth of edge, $e_{i,j} = 1$ indicates that the samples $x_i$ and $x_j$ belong to the same category ($y_i = y_j$); otherwise, $e_{i,j} = 0$. $\tilde{s}_{i,j}$ is the similarity score between $x_i$ and $x_j$ predicted by $f_{sim}$. Next, we update the parameters of $f_{agg}$ for feature aggregation using the cross-entropy (*ce*) as follows:

$$\mathcal{L}_{ce}^T(p_T(x_i), y_i) = -y_i\log\big(p_T(x_i)\big), \tag{5}$$

where $\mathcal{L}_{ce}^T$ denotes the cross-entropy loss function. $p_T(x_i) = \texttt{softmax}(z_i^T)$ indicates the prediction of $x_i$ with the aggregated features $z_i^T$ in Eq. (3), and $y_i \in [0,1]^C$ is the ground truth in one-hot encoding form.

We can easily adapt Eq. (4) and Eq. (5) on the labeled source $D_{src} = \{x_i^{src}, y_i^{src}\}_{i=1}^{N_{src}}$ to enrich intra-class relation of the source domain as follows.

$$\min_{\theta_{vit}, \theta_{sim}, \theta_{agg}} \mathcal{L}_{bce}^T(\tilde{s}_{i,j}^{src}, e_{i,j}^{src}) + \mathcal{L}_{ce}^T(p_T(x_i^{src}), y_i^{src}), \tag{6}$$

where $\mathcal{L}_{bce}^T(\tilde{s}_{i,j}^{src}, e_{i,j}^{src})$ and $\mathcal{L}_{ce}^T(p_T(x_i^{src}), y_i^{src})$ are the *bce* and *ce* losses used to update $\theta_{vit}$, $\theta_{sim}$, and $\theta_{agg}$ on the labeled source data, respectively. The proposed teacher network goes beyond obtaining the semantic features of individual images. Furthermore, it can comprehend the similarities between the neighboring samples, thus *enhancing intra-class consistency*. To enable intra-class relationships in the unlabeled target data, we assign pseudo-labels through a label generation process.

**Pseudo-Label Generation.** Following (Sohn et al., 2020), we first input the target image $x_i^{tar}$ into the teacher network, and the resulting prediction $p_T(x_i^{tar})$ is then converted into a one-hot hard label as follows:

$$\hat{y}_i^{ps} = \texttt{argmax}\big(p_T(x_i^{tar})\big) \text{ if } \texttt{max}\big(p_T(x_i^{tar})\big) \geq \tau, \tag{7}$$

where $\tau$ is a confidence threshold that controls the quality of the generated pseudo labels. Thanks to Eq. (7), we can obtain a pseudo-labeled set: $D_{ps} = \{(x_i^{tar}, \hat{y}_i^{ps})\}_{i=1}^{N_{ps}}$ from the unlabeled set $D_{tar} = \{(x_i^{tar})\}_{i=1}^{N_{tar}}$, where $N_{ps}$ denotes the number of pseudo labels and $N_{ps} \leq N_{tar}$.

**Self-Enhanced Teacher Learning.** In the next step, we combine $D_{ps} = \{(x_i^{tar}, \hat{y}_i^{ps})\}_{i=1}^{N_{ps}}$ into the source $D_{src} = \{(x_i^{src}, y_i^{src})\}_{i=1}^{N_{src}}$ as follows:

$$D_{cb} = D_{src} \cup D_{ps}, N_{cb} = N_{ps} + N_{src}, \tag{8}$$

where $N_{cb}$ is the number of combined samples. Since the combined dataset $D_{cb}$ consists of labeled data, it enables the use of supervised losses as described in Eq. (4) and Eq. (5). Therefore, Eq. (6) can be rewritten as follows:

$$\min_{\theta_{vit}, \theta_{sim}, \theta_{agg}} \mathcal{L}_{bce}^T(\tilde{s}_{i,j}^{cb}, e_{i,j}^{cb}) + \mathcal{L}_{ce}^T(p_T(x_i^{cb}), y_i^{cb}), \tag{9}$$

where $\tilde{s}_{i,j}^{cb}$ and $e_{i,j}^{cb}$ denote the similarity score and ground-truth edge of $\{(x_i^{cb}, y_i^{cb})\}_{i=1}^{N_{cb}} \in D_{cb}$, determined similarly to Equation Eq. (4). Notably, the teacher model enriches semantic representations and alleviates domain discrepancy when trained on $D_{cb}$, as it preserves intra-class relations in the source domain where $y_i^{src} = y_j^{src}$ within $D_{cb}$. Besides, it also leverages pseudo labels to exploit the intra-class relation of the target domain when $\hat{y}_i^{ps} = \hat{y}_j^{ps}$. Moreover, our teacher network addresses domain shift by *class-aware feature alignment* when $D_{cb}$ includes pairs with $\hat{y}_i^{ps} = y_j^{src}$.

Furthermore, we minimize cross-class confusion (MCC) (Jin et al., 2020) on $D_{tar}$ to enhance the pseudo-label generation process of the teacher network as follows:

$$\mathcal{L}_{cc}(p_T(x_i^{tar})) = \frac{1}{C}\sum_{c=1}^C\sum_{c'\neq c}^C |(p_T(x_{i,c}^{tar})^\top(p_T(x_{i,c'}^{tar})|, \tag{10}$$

where $p_T(x_{i,c}^{tar})$ and $p_T(x_{i,c'}^{tar})$ represent the probabilities of the target sample $x_i^{tar}$ belonging to the $c$-th and $c'$-th classes, respectively, where $\{c, c'\} \in C$. $\mathcal{L}_{cc}(p_T(x_i^{tar}))$ is minimized to alleviate the cross-class confusion level between the $c$-th and $c'$-th classes of the target samples $x_i^{tar}$.

## 3.5 TRAINING STRATEGY FOR STUDENT NETWORK

We also train the student network with the combined dataset $D_{cb} = \{(x_i^{cb}, y_i^{cb})\}_{i=1}^{N_{cb}}$ using the cross-entropy loss as:

$$\mathcal{L}_{ce}^S(p_S(x_i^{cb}), y_i^{cb}) = -y_i^{cb} \log(p_S(x_i^{cb})). \tag{11}$$

As defined in Eq. (8), $D_{cb}$ consists of both $D_{src}$ and $D_{ps}$. Thus, the student network can *Obtain Source Knowledge* on labeled source samples and effectively learn on unlabeled target data via *Pseudo Label-Based Gradient Guidance*.

**Obtain Source Knowledge.** The student network captures the knowledge in $D_{src} = \{(x_i^{src}, y_i^{src})\}_{i=1}^{N_{src}} \in D_{cb}$ using the cross-entropy loss as follows:

$$\mathcal{L}_{ce}^S(p_S(x_i^{src}), y_i^{src}) = -y_i^{src} \log(p_S(x_i^{src})), \tag{12}$$

where $p_S(x_i^{src}) = \texttt{softmax}(\text{MLP}(f_{cnn}(x_i^{src})))$ represents the output prediction of the source image $x_i^{src}$, which can be rewritten as follows:

$$\min_{\theta_{cnn}, \theta_{mlp}} \mathcal{L}_{ce}^S(p_S(x_i^{src}), y_i^{src}). \tag{13}$$

The student's parameters $\theta_S$ including $\theta_{cnn}$ and $\theta_{mlp}$.

**Pseudo Label-Based Gradient Guidance.** We use $D_{ps} = \{(x_i^{tar}, \hat{y}_i^{ps})\}_{i=1}^{N_{ps}} \in D_{cb}$ to adjust the gradient direction in updating the parameters of the student network as follows:

$$\theta_S \leftarrow \theta_S - \eta \nabla_{\theta_S} \mathcal{L}_{ce}^S(p_S(x_i^{tar}), \hat{y}_i^{ps}), \tag{14}$$

where $\nabla_{\theta_S}$ is the gradient of the loss $\mathcal{L}_{ce}^S$ with respect to $\theta_S$ on the unlabeled target data with the learning rate $\eta$. The student network provides prediction on the target image $x_i^{tar}$ by $p_S(x_i^{tar}) = \texttt{softmax}(\text{MLP}(f_{cnn}(x_i^{tar})))$. $\hat{y}_i^{ps}$ refers to the pseudo label generated by teacher network using Eq. (7), which guides the gradient $\nabla_{\theta_S} \mathcal{L}_{ce}^S(p_S(x_i^{tar}), \hat{y}_i^{ps})$, distilling the knowledge from the teacher network to the student network. The goal of the gradient guidance strategy is to align the student's gradient directions with those of the teacher, ensuring the student behaves similarly. Consequently, the student converges toward the optimal solution alongside its teacher on the target data, satisfying *the factor* ②, as verified by the qualitative visualization in the analysis section.

## 3.6 IMPLEMENTATION DETAILS

The training procedure of GraDA is processed in each episode $e$ consisting of a fixed number of training steps $t$. Specifically, in the initial episode ($e = 0$), $D_{src}$ is sampled into multiple mini-batches of size $B$ to facilitate training of both teacher and student networks as in Eq. (6) and Eq. (12), respectively. After completing the initial episode, the teacher network is utilized to generate the first pseudo-labeled set $D_{ps}^1$ from the unlabeled target data $D_{tar}$ for the next episode ($e = 1$) using Eq. (7). $D_{ps}^1$ is then combined with $D_{src}$ to form $D_{cb}^1 = D_{src} \cup D_{ps}^1$, as specified in Eq. (8), which is summarized as follows:

$$D_{cb}^e = D_{src} \cup D_{ps}^e, \text{ where } D_{cb}^0 = D_{src}. \tag{15}$$

Finally, $D_{cb}^e$ is divided into mini-batches of size $B$ to update the parameters of the teacher and student networks by using Eq. (9) and Eq. (11), respectively. This iterative process continues until convergence, where both networks align at an optimal point within a flattened region of the loss surface.

## 4 EXPERIMENTS

### 4.1 SETUP

**Dataset.** We conduct experiments on **VisDA2017** (Peng et al., 2018) with the domain adaptation task: *Synthetic* to *Real-world*. **Office-Home** (Venkateswara et al., 2017) includes 4 different domains: *Art* (Ar), *Clipart* (Cl), *Product* (Pr), and *Real-World* (Rw), providing 12 DA tasks. A subset

**(a) VisDA2017**

| Net | Method | Mean |
|---|---|---|
| ResNet101 | MCC (ECCV'20) | 78.8 |
| | STAR (CVPR'20) | 82.7 |
| | FixBi (CVPR'21) | 87.2 |
| | DAMP (CVPR'24) | 88.4 |
| | HVCLIP (ECCV'24) | 90.0 |
| | ☞GraDA (S) | 96.5 |
| ViT-B | PMTrans (CVPR'23) | 87.5 |
| | SSRT (CVPR'22) | 88.8 |
| | DAMP (CVPR'24) | 90.9 |
| | NVC (WACV'24) | 91.4 |
| | HVCLIP (ECCV'24) | 92.5 |
| | ☞GraDA (T) | 97.0 |

**(b) Office-Home**

| Net | Method | Ar→Cl | Ar→Pr | Ar→Rw | Cl→Ar | Cl→Pr | Cl→Rw | Pr→Ar | Pr→Cl | Pr→Rw | Rw→Ar | Rw→Cl | Rw→Pr | Mean |
|---|---|---|---|---|---|---|---|---|---|---|---|---|---|---|
| ResNet50 | SCDA (ICCV'21) | 57.5 | 76.9 | 80.3 | 65.7 | 74.9 | 74.5 | 65.5 | 53.6 | 79.8 | 74.5 | 59.6 | 83.7 | 70.5 |
| | DALN (CVPR'22) | 57.8 | 79.9 | 82.0 | 66.3 | 76.2 | 77.2 | 66.7 | 55.5 | 81.3 | 73.5 | 60.4 | 85.3 | 71.8 |
| | AML (IEEE Trans'23) | 58.9 | 77.2 | 81.7 | 69.6 | 77.9 | 78.6 | 66.6 | 57.9 | 82.3 | 74.7 | 62.5 | 84.5 | 72.7 |
| | GeT (ICCV'23) | 59.4 | 79.6 | 82.9 | 71.4 | 79.8 | 79.8 | 69.7 | 56.2 | 83.5 | 73.9 | 60.1 | 86.0 | 73.5 |
| | DAMP (CVPR'24) | 59.7 | 88.5 | 86.8 | 76.6 | 88.9 | 87.0 | 76.3 | 59.6 | 87.1 | 77.0 | 61.0 | 89.9 | 78.2 |
| | HVCLIP (ECCV'24) | 62.0 | 85.8 | 86.2 | 77.8 | 84.3 | 86.8 | 80.7 | 66.5 | 87.8 | 80.3 | 64.9 | 90.4 | 79.5 |
| | ☞GraDA (S) | **88.6** | **94.8** | **97.0** | **93.9** | **93.7** | **96.0** | **92.7** | **88.3** | **97.2** | **95.0** | **90.9** | **97.6** | **93.8** |
| ViT-B | TVT (WACV'23) | 74.9 | 86.8 | 89.5 | 82.8 | 88.0 | 88.3 | 79.8 | 71.9 | 90.1 | 85.5 | 74.6 | 90.6 | 83.6 |
| | SSRT (CVPR'22) | 75.2 | 89.0 | 91.1 | 85.1 | 88.3 | 90.0 | 85.0 | 74.2 | 91.3 | 85.7 | 78.6 | 91.8 | 85.4 |
| | NVC (WACV'24) | 75.1 | 89.0 | 91.5 | 86.4 | 88.6 | 90.2 | 84.8 | 73.7 | 91.7 | 87.1 | 74.6 | 92.9 | 85.5 |
| | DAMP (CVPR'24) | 75.7 | 94.2 | 92.0 | 86.3 | 94.2 | 91.9 | 86.2 | 76.3 | 92.4 | 86.1 | 75.6 | 94.0 | 87.1 |
| | PMTrans (CVPR'23) | 81.2 | 91.6 | 92.4 | 88.9 | 91.6 | 93.0 | 88.5 | 80.0 | 93.4 | 89.5 | 82.4 | 94.5 | 88.9 |
| | HVCLIP (ECCV'24) | 86.3 | **96.4** | 94.0 | 91.6 | **97.9** | 94.6 | 87.5 | 85.3 | 94.8 | 89.9 | 88.1 | 97.0 | 92.0 |
| | ☞GraDA (T) | **89.3** | 94.8 | **97.2** | **94.1** | 93.8 | **96.2** | **92.8** | **89.1** | **97.4** | **95.1** | **91.5** | **97.7** | **94.0** |

Table 1: Accuracy (%) on (a) **VisDA2017** and (b) **Office-Home** under the UDA setting. We compare the results of the student GraDA (S) to previous CNN-based works for fairness, while the comparison of the teacher GraDA (T) and ViT-based DA works is provided for reference. The best results are marked as **bold**. For **VisDA2017**, the per-class accuracy is in the *Suppl. Material*.

| Net | Method | rel→clp | | rel→pnt | | pnt→clp | | clp→skt | | skt→pnt | | rel→skt | | pnt→rel | | Mean | |
|---|---|---|---|---|---|---|---|---|---|---|---|---|---|---|---|---|---|
| | | 1-shot | 3-shot | 1-shot | 3-shot | 1-shot | 3-shot | 1-shot | 3-shot | 1-shot | 3-shot | 1-shot | 3-shot | 1-shot | 3-shot | 1-shot | 3-shot |
| ResNet34 | MME (ICCV'19) | 70.0 | 72.2 | 67.7 | 69.7 | 69.0 | 71.7 | 56.3 | 61.8 | 64.8 | 66.8 | 61.0 | 61.9 | 76.1 | 78.5 | 66.4 | 68.9 |
| | APE (ECCV'20) | 70.4 | 76.6 | 70.8 | 72.1 | 72.9 | 76.7 | 56.7 | 63.1 | 64.5 | 66.1 | 63.0 | 67.8 | 76.6 | 79.4 | 67.6 | 71.7 |
| | SPA (NIPS'23) | 75.3 | 76.0 | 71.8 | 72.2 | 74.8 | 76.5 | 65.9 | 67.0 | 69.8 | 71.1 | 65.8 | 67.2 | 81.1 | 82.3 | 72.1 | 73.2 |
| | GeT (ICCV'23) | 76.1 | 77.6 | 72.5 | 73.9 | 73.9 | 75.8 | 66.7 | 67.8 | 69.8 | 73.6 | 66.8 | 67.1 | 82.0 | 82.8 | 72.2 | 73.9 |
| | DECOTA (ICCV'21) | 79.1 | 80.4 | 74.9 | 75.2 | 76.9 | 78.7 | 65.1 | 68.6 | 72.0 | 72.7 | 69.7 | 71.9 | 79.6 | 81.5 | 73.9 | 75.6 |
| | CDAC (CVPR'21) | 77.4 | 79.6 | 74.2 | 75.1 | 75.5 | 79.3 | 67.6 | 69.9 | 71.0 | 73.4 | 69.2 | 72.5 | 80.4 | 81.9 | 73.6 | 76.0 |
| | ECACL (ICCV'21) | 75.3 | 79.0 | 74.1 | 77.3 | 75.3 | 79.4 | 65.0 | 70.6 | 72.1 | 74.6 | 68.1 | 71.6 | 79.7 | 82.4 | 72.8 | 76.4 |
| | MCL (IJCAI'22) | 77.4 | 79.4 | 74.6 | 76.3 | 75.5 | 78.8 | 66.4 | 70.9 | 74.0 | 74.7 | 70.7 | 72.3 | 82.0 | 83.3 | 74.4 | 76.5 |
| | SLA (CVPR'23) | 79.8 | 81.6 | 75.6 | 76.0 | 77.4 | 80.3 | 68.1 | 71.3 | 71.7 | 73.5 | 71.7 | 73.5 | 80.4 | 82.5 | 75.0 | 76.9 |
| | EFTL (AAAI'24) | 79.6 | 81.2 | 74.9 | 77.1 | 78.2 | 81.8 | 69.3 | 71.8 | 71.8 | 74.4 | 69.9 | 71.5 | 83.1 | 84.4 | 75.3 | 77.6 |
| | FMLM (ECCV'24) | 80.9 | 81.1 | 79.9 | 80.2 | 80.1 | 81.1 | 73.7 | 76.8 | 79.2 | 82.5 | 78.4 | 78.5 | 86.9 | 90.1 | 78.7 | 81.2 |
| | ☞GraDA (S) | **94.5** | **96.3** | **96.6** | **97.0** | **95.3** | **95.5** | **91.5** | **93.5** | **95.5** | **95.9** | **93.6** | **93.8** | **95.3** | **96.5** | **94.8** | **95.5** |
| ViT-B | ☞GraDA (T) | 95.2 | 97.0 | 97.1 | 97.7 | 96.0 | 96.2 | 92.1 | 94.2 | 96.1 | 96.5 | 94.4 | 94.5 | 96.6 | 97.3 | 95.4 | 96.2 |

Table 2: Accuracy (%) on **DomainNet** under the SSDA setting. The best results are marked as **bold**.

of **DomainNet** (Peng et al., 2019) includes 126 classes in 4 diverse domains: *real* (rel), *clipart* (clp), *painting* (pnt), and *sketch* (skt), where we follow previous works (Saito et al., 2019; Zhang & Lee, 2023) to verify our method on 7 DA tasks. More information is provided in the *Suppl. Material*.

**Experimental Settings.** All experiments were conducted on a single RTX-4090 GPU. For the feature extractor of the teacher network, we utilized ViT-B model with a $16 \times 16$ patch size, whereas the ResNet family (He et al., 2016) served as the feature extractor for the student network, similar to (Lu et al., 2020; Xiao et al., 2023). Each feature extractor was pre-trained on ImageNet-1k. For the classifier, GCN (Luo et al., 2020) was used to implement the CA module in the teacher network, while a two-layer MLP (Li et al., 2021a; Yu & Lin, 2023) was employed in the student network. Both networks were optimized using SGD with a learning rate and weight decay of $5 \times 10^{-4}$, and momentum of 0.9, respectively. We set the mini-batch size to $B = 32$ and the pseudo-label threshold in Eq. (7) to $\tau = 0.95$. Teacher and student networks are trained for $E = 100$ episodes, with $T = 500$ steps for **Office-Home** and $T = 1,000$ for **VisDA2017** and **DomainNet** per episode.

## 4.2 COMPARISON WITH STATE-OF-THE-ARTS

Notably, the results of the student serve as a baseline for a fair comparison with prior CNN-based methods, while the results of the teacher are used solely for analysis.

**UDA Methods.** Tables 1a and 1b show the results on **VisDA2017** and **Office-Home**, respectively. Specifically, on **VisDA2017**, GraDA (S) and GraDA (T) achieve 96.5% and 97.0%, respectively, indicating a minimal performance gap between the student and teacher networks. Additionally, GraDA (S) outperforms the second-best method, HVCLIP Vesdapunt et al. (2024), with a gain of 6.5%. Under a fair comparison using the same ViT-B backbone, GraDA (T) also surpasses HVCLIP by 5.5%. On **Office-Home**, GraDA (S) still achieves the best results across all tasks, surpassing the DA method with KD, AML Zhou et al. (2023), in several challenging tasks, such as Ar→Cl, Pr→Cl, and Rw→Cl, with notable accuracies of 88.6%, 88.3%, and 90.9%, respectively. As a result, the mean accuracy of GraDA (S) reaches 93.8% across 12 DA tasks, improving by 14.3% compared to the second-best method, HVCLIP. Mean accuracy of GraDA (T) also achieves 94.0%, it surpasses all ViT-based competitors, and exceeds the second-best HVCLIP by 2.0%.

| (a) *Art→Clipart* (UDA) | (b) *Product→Art* (UDA) | (c) *skt→pnt* (1-shot) | (d) *rel→skt* (3-shot) |

Figure 3: Convergence trajectory in the loss landscape of teacher and student networks.

| Scenario | | Settings | Office-Home (UDA) (ResNet50) | DomainNet (SSDA) (ResNet34) |
|---|---|---|---|---|
| | | | **Student** Mean Acc. (%) | |
| Vanilla **Student** | S1 | Supervised | 59.4 | 60.0 |
| | S2 | + Self-Enhanced | 62.7 | 71.3 |
| | S3 | + $\mathcal{L}_{cc}^{S}(p_S(x_i^{tar}))$ | 69.1 | 73.4 |
| With **Teacher** | S4 | Supervised | 84.0 | 87.1 |
| | S5 | + Self-Enhanced | 93.5 | 95.0 |
| | S6 | + $\mathcal{L}_{cc}^{T}(p_T(x_i^{tar}))$ | 93.8 | 95.5 |

Table 3: Ablation study on **Office-Home** and **DomainNet** under UDA and 3-shot SSDA.

| Teacher-Student Pair | | | rel→clp | clp→skt | skt→pnt | pnt→rel | Mean |
|---|---|---|---|---|---|---|---|
| **P1** | **T** | ResNet101+MLP | 77.0 | 68.1 | 73.6 | 81.7 | 75.1 |
| | **S** | ResNet34+MLP | 76.8 | 67.6 | 73.3 | 81.3 | 74.8 |
| **P2** | **T** | ViT-B+MLP | 85.3 | 79.1 | 85.3 | 90.9 | 85.2 |
| | **S** | ResNet34+MLP | 85.3 | 78.6 | 85.1 | 89.6 | 84.7 |
| **P3** | **T** | ResNet101+CA | 95.1 | 90.5 | 93.0 | 91.8 | 92.6 |
| | **S** | ResNet34+MLP | 93.9 | 88.6 | 90.9 | 93.9 | 91.8 |
| **P4** | **T** | ViT-B+CA | 97.0 | 94.2 | 96.5 | 97.3 | 96.3 |
| | **S** | ResNet34+MLP | 96.3 | 93.5 | 95.9 | 96.5 | 95.6 |

Table 4: Performance of student (**S**) paired with various teachers (**T**) on **DomainNet** (3-shot).

**SSDA Methods.** Similar to Saito et al. (2019); Li et al. (2021a), we simply add a few labeled target samples (1-shot or 3-shot) into the training dataset under the SSDA setting. As listed in Table 2, GraDA (**S**) provides the remarkable results across 7 DA tasks on **DomainNet** with an average accuracy of 94.8% and 95.5% corresponding to the 1-shot and 3-shot settings, respectively. Moreover, the average accuracy gap between GraDA (**S**) and GraDA (**T**) is narrowed to 0.6% and 0.7% under the 1-shot and 3-shot, respectively. These results demonstrate that the student network, utilizing a *small* model (ResNet34), can efficiently capture the knowledge of the *larger* teacher network (ViT-B).

## 4.3 ANALYSES

As observed, in settings **S1**, **S2**, and **S3**, where the student network operates without guidance from the teacher network, it achieves maximum accuracies of only 69.1% and 73.4% on **Office-Home** and **DomainNet**, respectively. In contrast, with teacher guidance in settings **S4**, **S5**, and **S6**, the classification performance of the student network is significantly enhanced. Specifically, in setting **S4**, the student network's results improve by 14.9% and 13.7%, despite the limited quality and quantity of pseudo la-

| Teacher-Student Pair | #Params (M) | Ar→Cl | Cl→Pr | Pr→Rw | Rw→Ar | Mean |
|---|---|---|---|---|---|---|
| ResNet50+CA (**T**) | 30.3 | 78.3 | 83.9 | 93.8 | 92.5 | 87.1 |
| ResNet50+MLP (**S**) | 24.6 | 77.8 | 83.8 | 93.5 | 92.3 | 86.9 |
| ViT-tiny+CA (**T**) | 8.1 | 68.7 | 81.4 | 93.0 | 87.5 | 82.7 |
| ViT-tiny+MLP (**S**) | 5.7 | 68.3 | 80.5 | 92.8 | 87.3 | 82.2 |
| HVCLIP (ResNet50) | ≈101.5 | 62.0 | 84.3 | 87.8 | 80.3 | 78.6 |

Table 5: Ablation study on various teacher-student pairs on **Office-Home** with UDA. (*Complete DA tasks in Suppl.*)

bels provided by the teacher network, as only intra-class relationships within the source domain are considered in Eq. (6). In setting **S5**, the teacher network improves generalization to unlabeled target data thanks to intra-class relationships, while class-aware feature alignment mitigates the domain shift issue using Eq. (9). Furthermore, setting **S6** overcomes the ambiguous class confusion using Eq. (10). As a result, the quality and quantity of pseudo labels from the teacher network increase, enhancing the student network's performance.

**Can a Teacher Truly Educate a Student?** To examine this, we use the gradient trajectory to observe the changes in the learning behavior of teacher and student networks with the pseudo label-based gradient guidance algorithm. We visualize the convergence trajectory of two UDA tasks on **Office-Home**: Ar→Cl and Pr→Ar (Fig. 3a), two SSDA tasks on **DomainNet**: skt→pnt and rel→skt with 1-shot and 3-shot settings (Fig. 3b), respectively. As shown in these figures, both the teacher and student models are initialized with random parameters, leading to different starting points. Nevertheless, the teacher network converges toward an optimal solution, followed by the student, ultimately aligning within a minimal region with a low loss value.

**Can a Student Perform Better with a Better Teacher?** We verify the critical role of teacher network design in optimizing the effectiveness of the knowledge distillation scheme. Experiments are conducted using ResNet34+MLP as the anchor student network, paired with various types of teacher networks such as ResNet101+MLP, ViT-B+MLP, ResNet101+CA, and ViT-B+CA corresponding to **P1**, **P2**, **P3**, and **P4**, respectively. These teacher-student pairs are evaluated on 4 DA tasks within **DomainNet** under a 3-shot setting. As reported in Tab. 4, the results of **P2** significantly outperform those of **P1**, attributed to ViT-B's superior capacity for image representation compared to ResNet101. However, a comparison among **P1**, **P2**, and **P3** reveals a critical insight: the category-level aggregation (+CA) module proves to be a pivotal component in enhancing the effectiveness of knowledge distillation. The CA module not only facilitates improved intra-class generalization within source and target domains but also mitigates domain shift across these domains through class-aware feature alignment. Based on these findings, ViT-B+CA, associated with **P4**, is selected as the optimal teacher network, surpassing the

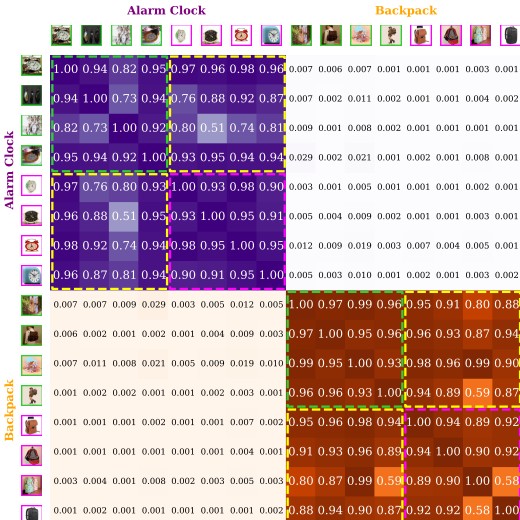

Figure 4: Illustration of category-level aggregation (similarity matrix $\tilde{S}$) of the UDA task Ar→Rw on **Office-Home**. Relationships between samples within the source and target domains are outlined in dashed green and pink boxes, respectively. Relationships of cross-domain samples are outlined by dashed yellow boxes.

other teacher networks and achieving the best accuracy of 96.3%. The classification performance of the student network in **P4** exceeds that of the student network in **P1**, including the least effective teacher network, by 20.8%.

**Fairness of the Teacher Network.** We conduct an ablation where the (**T**) and (**S**) networks share the same backbone on four **Office-Home** UDA tasks (Tab. 5). Even with identical ResNet50 or the smaller ViT-Tiny, GraDA still outperforms HVCLIP (Tab. 1b). This demonstrates that GraDA's gains stem from the proposed CA module and effective pseudo labels, not just the backbone.

**Effectiveness of CA Module.** The category-level aggregation (CA) mechanism plays a crucial role in the teacher network, which enhances *intra-class representations* within a domain and facilitates *class-aware alignment* across domains. To demonstrate this, we present Fig. 4, which visualizes the similarity matrix $\tilde{S}$ generated by the teacher network for a mini-batch ($B = 16$) in the Ar→Rw task, encompassing both source and target test samples. As shown, CA functions effectively, exhibiting high similarity among same-category samples, both within and across domains.

## 5 CONCLUSION

We introduce GraDA, a novel method designed to enhance student network performance in knowledge distillation for domain adaptation tasks. To achieve this, we first developed a strong teacher network by integrating a ViT backbone with a Category-level Aggregation (CA) module to produce robust representations. The CA module enhances the teacher's generalization ability by capturing *intra-class relations* within each domain and reducing domain shift between domains through *class-aware feature alignment*. We then proposed a gradient-guided knowledge distillation approach to optimize the transfer of knowledge from the ViT-based teacher to a lightweight CNN-based student, which is primarily used during inference. By providing high-quality pseudo labels, the ViT-based teacher guides the gradient updates of the student's parameters. Experiments across diverse settings demonstrate that GraDA significantly outperforms state-of-the-art methods on widely used benchmarks. Notably, this success is fully explainable, as evidenced by thorough qualitative analyses.

**Limitation.** While the CA module improves the teacher's generalization, aggregating features from noisy labels may introduce accumulated errors, degrading its quality and guidance to the student. It would be interesting to investigate incorporating a denoising module before the CA module.

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

# A  APPENDIX

This *Supplementary Material* provides five-fold information. First, we summarize the notations frequently used in the main manuscript and their corresponding definitions (Sec. B) and a detailed overview of all datasets (Sec. C). Second, a concise pseudocode for GraDA is included for clarity (Sec. D), and the implementation details of the teacher and student networks are thoroughly explained (Sec. E). Third, additional results in the UDA and SSDA settings are presented in Sec. F. Fourth, various aspects are discussed in more detail, including the diversity of student networks in GraDA, the pivotal role of CA in teacher network design, and the influence of CA on student performance through pseudo-label generation, supported by additional results and discussion (Sec. G). Finally, qualitative results using t-SNE and Grad-CAM are provided to further visually evaluate GraDA (Sec. H). Below, we present the table of contents to facilitate easy access to the information.

# Contents

| | Notation | Definition |
|---|---|---|
| **Abbreviation** | DA | Domain Adaptation |
| | UDA | Unsupervised Domain Adaptation |
| | SSDA | Semi-supervised Domain Adaptation |
| | KD | Knowledge Distillation |
| | CNN | Convolutional Neural Networks |
| | ViT | Vision Transformer |
| | MLP | Multilayer Perceptron |
| | CA | Category-level Aggregation |
| | GCN | Graph Convolutional Networks |
| **Symbol in Data Setting** | $D_{src}$ | Set of the labeled source domain |
| | $x_i^{src}$ | The $i$-th image from the source domain |
| | $y_i^{src}$ | Label of the source image $x_i^{src}$ |
| | $N_{src}$ | The number of samples in the source domain |
| | $D_{tar}$ | Set of the unlabeled target domain |
| | $x_i^{tar}$ | The $i$-th image from the target domain |
| | $y_i^{tar}$ | Label of the target image $x_i^{tar}$ |
| | $N_{tar}$ | The number of samples in the target dataset |
| | $C$ | Number of categories in both domains |
| | $D_{ps}^e$ | Pseudo-labeled set generated at episode $e$ |
| | $\hat{y}_i^{ps}$ | Pseudo label for the target image $x_i^{tar}$ |
| | $N_{ps}$ | The number of samples in the pseudo-labeled set |
| | $D_{cb}^e$ | Combined dataset at episode $e$ of $D_{src}$ and $D_{ps}^e$ |
| | $x_i^{cb}$ | The $i$-th image from the combined set |
| | $y_i^{cb}$ | Label of the combined image $x_i^{cb}$ |
| | $N_{cb}$ | The number of samples in the combined set |
| **Symbol in Train.** | $\eta$ | Learning rate of the teacher and student networks |
| | $\tau$ | Confidence threshold for pseudo-label generation |
| | $B$ | The number of samples in a mini-batch |
| | $E$ | The number of training episodes |
| | $e$ | Episode index in the training process |
| | $T$ | Number of training steps per episode |

| | Notation | Definition |
|---|---|---|
| **Symbol in Teacher** | $f_{vit}$ | ViT-based feature extractor |
| | $f_{sim}$ | Similarity network in the CA module |
| | $f_{agg}$ | Category aggregation network in the CA module |
| | $\theta_{vit}$ | Set of learnable parameters for $f_{vit}$ |
| | $\theta_{sim}$ | Set of learnable parameters for $f_{sim}$ |
| | $\theta_{agg}$ | Set of learnable parameters for $f_{agg}$ |
| | $z_i^{vit}$ | Feature vector generated by $f_{vit}$ for sample $x_i$ |
| | $d$ | Embedding size of the feature vector $z_i^{vit}$ |
| | $\hat{s}_{i,j}$ | Similarity score between feature vectors |
| | $\hat{S}$ | Similarity matrix within a mini-batch $B$ |
| | $\tilde{s}_{i,j}$ | Normalized similarity score |
| | $\tilde{S}$ | Normalized similarity matrix |
| | $I$ | Identity matrix used for normalization |
| | $D$ | Degree matrix of $\hat{S} + I$ |
| | $z_i^T$ | Aggregated feature vector |
| | $p_T(x_i)$ | Prediction of the teacher network for sample $x_i$ |
| | $e_{i,j}$ | Ground-truth of edge between two samples |
| | $\mathcal{L}_{bce}^T$ | Binary cross-entropy loss of the teacher network |
| | $\mathcal{L}_{ce}^T$ | Cross-entropy loss of the teacher network |
| | $\mathcal{L}_{cc}$ | Cross-class confusion loss of the teacher network |
| **Symbol in Student** | $f_{cnn}$ | CNN-based feature extractor |
| | MLP | MLP classifier |
| | $\theta_{cnn}$ | Set of learnable parameters for $f_{cnn}$ |
| | $\theta_{mlp}$ | Set of learnable parameters for the MLP |
| | $z_i^{cnn}$ | Feature vector generated by $f_{cnn}$ for sample $x_i$ |
| | $d'$ | Embedding size of the feature vector $z_i^{cnn}$ |
| | $p_S(x_i)$ | Prediction of the student network for sample $x_i$ |
| | $\mathcal{L}_{ce}^S$ | Cross-entropy loss of the student network |
| | $\nabla_{\theta_S}$ | The gradient of the loss $\mathcal{L}_{ce}^S$ with respect to $\theta_S$ |

Table 6: Abbreviation and symbol notation (**Train.** stands for Training).

## B  NOTATIONS

We summarize notations and their definitions frequently used in the proposed method, as listed in Tab. 6.

## C  DATASET DETAILS

Table 7 provides an overview of popular domain adaptation datasets, including **VisDA2017**, **Office-Home**, **DomainNet**, **Office-31**, and **ImageCLEF-DA**. It details the number of categories and the number of images for each dataset, along with sample images from different domains.

**VisDA2017** Peng et al. (2018) exhibits a significant domain gap when transferring from the *Synthetic* domain to the *Real-world* domain. It includes $152,397$ *Synthetic* images as the source domain and $55,388$ *Real-world* images as the target domain. Each domain consists of 12 different categories. The synthetic images are generated from 3D models, while the real images are collected from natural scenes.

**Office-Home** Venkateswara et al. (2017) contains approximately $15,500$ images across 65 categories from 4 distinct domains: *Art* (Ar), *Clipart* (Cl), *Product* (Pr), and *Real-World* (Rw). These 4 domains establish 12 cross-domain tasks: Ar→Cl, Ar→Pr, Ar→Rw, Cl→Ar, Cl→Pr, Cl→Rw, Pr→Ar, Pr→Cl, Pr→Rw, Rw→Ar, Rw→Cl, and Rw→Pr.

**DomainNet** Peng et al. (2019) contains approximately $600,000$ images from six domains: *clipart* (clp), *infograph* (inf), painting (pnt), *quickdraw* (qdw), *real* (rel), and *sketch* (skt). It includes 345 categories. In the SSDA setting, we use a subset with 126 classes across these domains to consistency with the prior SSDA works Saito et al. (2019); Qin et al. (2021); Yu & Lin (2023); Huang et al. (2023). In the UDA setting, we employ **Mini-DomainNet** as used in Prabhu et al. (2021); Westfechtel et al. (2023), a curated subset with 40 frequently observed classes across the same 4 domains, encompassing all 12 possible domain shifts.

**Algorithm 1:** Pseudocode of GraDA

**1 Input:** Source and target datasets: $D_{src} = \{(x_i^{src}, y_i^{src})\}_{i=1}^{N_{src}}$; $D_{tar} = \{x_i^{tar}\}_{i=1}^{N_{tar}}$;

**2 Training Configuration:** Threshold $\tau$, Total episodes $E$, training steps $T$, Mini-batch size $B$, Learning rate $\eta$

**3 Initialization:** Combined dataset $D_{cb}^0 \leftarrow D_{src}$

**4 Network Architectures:**

**5**      Teacher network: $f_{vit}$, $f_{sim}$ and $f_{agg}$. Set of parameters: $\theta_T = \{\theta_{vit}, \theta_{sim}, \theta_{agg}\}$

**6**      Student network: $f_{cnn}$ and MLP. Set of parameters: $\theta_S = \{\theta_{cnn}, \theta_{mlp}\}$;

**7** ▶ TRAINING: **for** $e = 1$ **to** $E$ **do**

**8**     **for** $t = 1$ **to** $T$ **do**

**9**        Sample $\{(x_i^{cb}, y_i^{cb})\}_{i=1}^{B} \in D_{cb}^e$

**10**        • Training Strategy for the Teacher Network:

**11**        $\{z_i^{vit}\}_{i=1}^{B} \leftarrow \{f_{vit}(x_i^{cb}; \theta_{vit})\}_{i=1}^{B}$

**12**        ◇ Initialize similarity matrix: $\hat{S} \in \mathbb{R}^{B \times B}$

**13**        **for** $(i, j)$ **in** $(1..B, 1..B)$ **do**

**14**           ◇ Computing the $(i, j)$-th similarity score:

**15**           $\hat{s}_{i,j} \leftarrow \texttt{sigmoid}\big(f_{sim}(\|z_i^{vit} - z_j^{vit}\|; \theta_{sim})\big)$ where $\hat{s}_{i,j} \in \hat{S}$      ▷ *Eq. (1).*

**16**        $\tilde{S} \leftarrow D^{-\frac{1}{2}}(\hat{S} + I)D^{-\frac{1}{2}}$      ▷ *Eq. (2).*

**17**        **for** $i = 1$ **to** $B$ **do**

**18**           $z_i^T \leftarrow f_{agg}\Big(\big[z_i^{vit}, \sum_{j \in B} \tilde{s}_{i,j} \cdot z_j^{vit}\big]; \theta_{agg}\Big)$

**19**        ▷ *Eq. (3).*

**20**           $p_T(x_i^{cb}) \leftarrow \texttt{softmax}(z_i^T)$

**21**        $\theta_T \leftarrow \theta_T - \eta \nabla_{\theta_T}(\mathcal{L}_{bce}^T + \mathcal{L}_{ce}^T + \mathcal{L}_{cc})$

**22**           ▷ $\mathcal{L}_{bce}^T$, $\mathcal{L}_{ce}^T$ *and* $\mathcal{L}_{cc}$ *are computed in Eq. (4), Eq. (5) and Eq. (10), respectively.*

**23**        • Training Strategy for Student Network: $z_i^{cnn} \leftarrow f_{cnn}(x_i^{cb}; \theta_{cnn})$

**24**        $p_S(x_i) \leftarrow \texttt{softmax}(\text{MLP}(z_i^{cnn}; \theta_{mlp}))$

**25**        $\theta_S \leftarrow \theta_S - \eta \nabla_{\theta_S} \mathcal{L}_{ce}^S$

**26**        ▷ $\mathcal{L}_{ce}^S$ *is computed in Eq. (11).*

**27**     ◇ Pseudo Label Generation: $D_{ps}^e \leftarrow \emptyset$

**28**     **for** $\{(x_i^{tar})\}_{i=1}^{B}$ **in** $D_{tar}$ **do**

**29**        $\{z_i^{vit}\}_{i=1}^{B} \leftarrow \{f_{vit}(x_i^{tar}; \theta_{vit})\}_{i=1}^{B}$

**30**        Repeat lines **13**–**20** to obtain $\{z_i^T\}_{i=1}^{B}$

**31**        **for** $i = 1$ **to** $B$ **do**

**32**           $p_T(x_i^{tar}) \leftarrow \texttt{softmax}(z_i^T)$

**33**           **if** $\max\big(p_T(x_i^{tar})\big) \geq \tau$ **then**

**34**              $\hat{y}_i^{ps} \leftarrow \texttt{argmax}\big(p_T(x_i^{tar})\big)$

**35**              ▷ *Assign pseudo label.*

**36**              $D_{ps}^e \leftarrow D_{ps}^e \cup \{(x_i^{tar}, \hat{y}_i^{ps})\}$

**37**              ▷ *Update pseudo-labeled set.*

**38**     ◇ Update the combined set: $D_{cb}^e \leftarrow D_{src} \cup D_{ps}^e$

**39** ▶ TESTING: $p_S(x_i^{tar}) \leftarrow \texttt{softmax}\big(\text{MLP}(f_{cnn}(x_i^{tar}))\big)$

**40**           ▷ *Only the student network is used for testing.*

**Office-31** Saenko et al. (2010) contains $4,110$ images across 31 categories from three distinct domains: *Amazon* (A), *Webcam* (W), and *DSLR* (D). The *Amazon* domain consists of images from online merchants, *Webcam* includes low-resolution images taken by web cameras, and *DSLR* contains high-resolution images captured with a digital SLR camera. In UDA, all 6 possible domain adaptation tasks between these domains are considered: A→W, A→D, W→A, W→D, D→A, and D→W. In the SSDA setting, 2 tasks are evaluated: W→A and D→A.

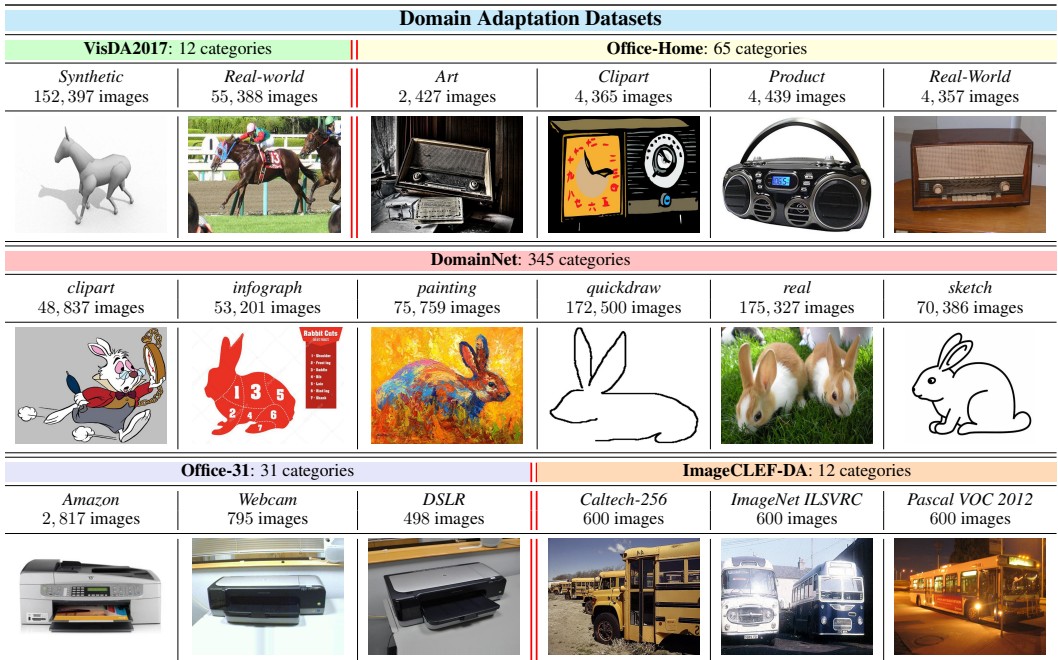

| Domain Adaptation Datasets | | | | | |
|---|---|---|---|---|---|
| **VisDA2017**: 12 categories | | **Office-Home**: 65 categories | | | |
| *Synthetic* 152, 397 images | *Real-world* 55, 388 images | *Art* 2, 427 images | *Clipart* 4, 365 images | *Product* 4, 439 images | *Real-World* 4, 357 images |
| **DomainNet**: 345 categories | | | | | |
| *clipart* 48, 837 images | *infograph* 53, 201 images | *painting* 75, 759 images | *quickdraw* 172, 500 images | *real* 175, 327 images | *sketch* 70, 386 images |
| **Office-31**: 31 categories | | | **ImageCLEF-DA**: 12 categories | | |
| *Amazon* 2, 817 images | *Webcam* 795 images | *DSLR* 498 images | *Caltech-256* 600 images | *ImageNet ILSVRC* 600 images | *Pascal VOC 2012* 600 images |

Table 7: Overview of popular domain adaptation datasets, including **VisDA2017**, **Office-Home**, **DomainNet**, **Office-31**, and **ImageCLEF-DA**. The number of images reflects the scale of each dataset, while the example images per domain highlight the distribution discrepancy.

| Net | Method | aero | bicycle | bus | car | horse | knife | motor | person | plant | skate | train | truck | Mean |
|---|---|---|---|---|---|---|---|---|---|---|---|---|---|---|
| ResNet101 | MCC (ECCV'20) | 88.1 | 80.3 | 80.5 | 71.5 | 90.1 | 93.2 | 85.0 | 71.6 | 89.4 | 73.8 | 85.0 | 36.9 | 78.8 |
| | STAR (CVPR'20) | 95.0 | 84.0 | 84.6 | 73.0 | 91.6 | 91.8 | 85.9 | 78.4 | 94.4 | 84.7 | 87.0 | 42.2 | 82.7 |
| | FixBi (CVPR'21) | 96.1 | 87.8 | 90.5 | **90.3** | 96.8 | 95.3 | 92.8 | 88.7 | 97.2 | 94.2 | 90.9 | 25.7 | 87.2 |
| | DAMP (CVPR'24) | 97.3 | 91.6 | 89.1 | 76.4 | 97.5 | 94.0 | 92.3 | 84.5 | 91.2 | 88.1 | 91.2 | 67.0 | 88.4 |
| | HVCLIP (ECCV'24) | 98.8 | 90.1 | 90.8 | 82.2 | 97.3 | 95.5 | 91.8 | 82.9 | 94.9 | 92.8 | 92.2 | 70.8 | 90.0 |
| | ☞ GraDA (**S**) | **99.9** | **98.6** | **96.4** | 88.4 | **100.0** | **99.8** | **99.3** | **97.5** | **100.0** | **100.0** | **99.2** | **78.7** | **96.5** |
| ViT-B | PMTrans (CVPR'23) | 98.9 | 93.7 | 84.5 | 73.3 | 99.0 | 98.0 | 96.2 | 67.8 | 94.2 | 98.4 | 96.6 | 49.0 | 87.5 |
| | SSRT (CVPR'22) | 98.9 | 87.6 | 89.1 | 84.8 | 98.3 | 98.7 | 96.3 | 81.1 | 94.9 | 97.9 | 94.5 | 43.1 | 88.8 |
| | DAMP (CVPR'24) | 98.7 | 92.8 | 91.7 | 80.1 | 98.9 | 96.9 | 94.9 | 83.2 | 93.9 | 94.9 | 94.8 | 70.2 | 90.9 |
| | NVC (WACV'24) | 98.5 | 89.0 | 88.5 | 92.0 | 98.5 | 98.3 | 96.2 | 88.4 | 98.5 | 97.9 | 95.0 | 55.4 | 91.4 |
| | HVCLIP (ECCV'24) | 99.0 | 93.7 | 92.1 | 84.5 | 98.8 | 96.2 | 94.2 | 88.6 | 96.9 | 96.7 | 94.5 | 74.4 | 92.5 |
| | ☞ GraDA (**T**) | **100.0** | **99.1** | **97.9** | **89.4** | **100.0** | **100.0** | **99.7** | **98.5** | **100.0** | **99.9** | **99.9** | **79.5** | **97.0** |

Table 8: Accuracy (%) on **VisDA2017** under the UDA setting. GraDA (**S**) and GraDA (**T**) are the student and teacher networks, respectively. For a fair comparison, we use GraDA (**S**) to compare with the prior CNN-based works, while the comparison of the teacher GraDA (**T**) and ViT-based DA works is provided for reference. The best classification accuracy is marked as **bold**.

**ImageCLEF-DA** Caputo et al. (2014) includes images from three domains: *Caltech-256* (C), *ImageNet ILSVRC 2012* (I), and *Pascal VOC 2012* (P). Each domain contains 12 categories with 50 images per category, totaling 600 images per domain. The dataset defines 6 domain adaptation tasks between these domains: I→P, P→I, I→C, C→I, C→P, and P→C.

## D PSEUDOCODE OF GRADA

We provide the pseudocode of GraDA presented in Algorithm 1, which is straightforward and helps to gain a better understanding of GraDA. Note that the losses for the teacher network including $\mathcal{L}_{bce}^{T}$, $\mathcal{L}_{ce}^{T}$, and $\mathcal{L}_{cc}$, are specified in Eqs. 4, 5 and 10, respectively. Meanwhile, the loss $\mathcal{L}_{ce}^{S}$ for the student network is detailed in Eq. 11.

| Net | Method | rel→clp | rel→pnt | rel→skt | clp→rel | clp→pnt | clp→skt | pnt→rel | pnt→clp | pnt→skt | skt→rel | skt→clp | skt→pnt | Mean |
|-----|--------|---------|---------|---------|---------|---------|---------|---------|---------|---------|---------|---------|---------|------|
| ResNet50 | MCD (CVPR'18) | 62.0 | 69.3 | 56.3 | 79.8 | 56.6 | 53.7 | 83.4 | 58.3 | 61.0 | 81.7 | 56.3 | 66.8 | 65.4 |
| | PADA (ECCV'18) | 65.9 | 67.1 | 58.4 | 74.7 | 53.1 | 52.9 | 79.8 | 59.3 | 57.9 | 76.5 | 67.0 | 61.1 | 64.5 |
| | BIWAA-I (WACV'23) | 79.9 | 75.2 | 75.4 | 87.9 | 72.1 | 75.7 | 88.9 | 77.8 | 76.7 | 88.8 | 80.5 | 74.5 | 79.4 |
| | SENTRY (ICCV'21) | 83.9 | 76.7 | 74.4 | 90.6 | 76.0 | 79.5 | 90.3 | 82.9 | 75.6 | 90.4 | 82.4 | 74.0 | 81.4 |
| | LUHP (AAAI'24) | 79.6 | 82.8 | 79.3 | 91.1 | 79.7 | 76.5 | 90.2 | 77.2 | 76.7 | 91.2 | 80.3 | 79.5 | 82.0 |
| | GSDE (WACV'24) | 82.9 | 79.2 | 80.8 | 91.9 | 78.2 | 80.0 | 90.9 | 84.1 | 79.2 | 90.3 | 83.4 | 76.1 | 83.1 |
| | ECB (CVPR'24) | 84.7 | 83.8 | 79.7 | 91.6 | 84.0 | 82.5 | 91.0 | 83.2 | 79.2 | 86.1 | 82.9 | 81.6 | 84.2 |
| | ☞ GraDA (S) | **93.1** | **94.8** | **85.7** | **98.4** | **95.0** | **91.5** | **97.3** | **86.6** | **89.3** | **95.7** | **93.1** | **96.4** | **93.1** |
| ViT-B | ☞ GraDA (T) | 93.4 | 95.0 | 86.2 | 98.7 | 95.5 | 91.9 | 98.2 | 86.9 | 90.0 | 96.4 | 93.4 | 96.7 | 93.5 |

Table 9: Accuracy (%) on **Mini-DomainNet** under the UDA setting. The best classification accuracy is marked as **bold**.

| Net | Method | A→W | D→W | W→D | A→D | D→A | W→A | Mean |
|-----|--------|-----|-----|-----|-----|-----|-----|------|
| ResNet50 | GVB-GD (CVPR'20) | 94.8 | 98.7 | 100.0 | 95.0 | 73.4 | 73.7 | 89.3 |
| | SCDA (ICCV'21) | 94.2 | 98.7 | 99.8 | 95.2 | 75.7 | 76.2 | 90.0 |
| | DALN (CVPR'22) | 95.2 | 99.1 | 100.0 | 95.4 | 76.4 | 76.5 | 90.4 |
| | BIWAA-I (WACV'23) | 95.6 | 99.0 | 100.0 | 94.4 | 75.9 | 77.3 | 90.5 |
| | GeT (ICCV'23) | 95.4 | 99.1 | 100.0 | 95.4 | 76.6 | 77.0 | 90.6 |
| | LUHP (AAAI'24) | 94.2 | 98.6 | 100.0 | 95.2 | 77.7 | 78.6 | 90.7 |
| | FixBi (CVPR'21) | 96.1 | 99.3 | 100.0 | 95.0 | 78.7 | 79.4 | 91.4 |
| | SPA (NIPS'23) | 97.2 | 99.0 | 99.8 | 95.0 | 78.0 | 79.4 | 91.4 |
| | HVCLIP (ECCV'24) | 96.2 | **99.4** | 100.0 | 96.0 | 80.1 | 80.6 | 92.1 |
| | ☞ GraDA (S) | **98.6** | 99.3 | **100.0** | **99.2** | **90.0** | **91.8** | **96.5** |
| ResNet34 | ☞ GraDA (S) | 99.3 | 99.4 | 100.0 | 99.2 | 90.8 | 91.4 | 96.7 |
| ResNet18 | ☞ GraDA (S) | 98.5 | 99.3 | 100.0 | 98.8 | 90.0 | 91.3 | 96.3 |
| ViT-B | ☞ GraDA (T) | 99.4 | 100.0 | 100.0 | 99.2 | 90.7 | 92.1 | 96.9 |

Table 10: Accuracy (%) on **Office-31** under the UDA setting with various versions of ResNet, such as ResNet50, ResNet34, and ResNet18. The best classification accuracy is marked as **bold**.

# E GraDA Architecture

This section thoroughly provides detailed implementations of teacher and student network architectures.

## E.1 Teacher Network

**Feature Extractor.** ViT-B Dosovitskiy et al. (2021) is adopted as the feature extractor $f_{vit}$, dividing the input image $x_i$ into patches of size $16 \times 16$. After processing by the patch embedding network, a sequence of 144 patch tokens is obtained, with the [CLS] token added at the beginning. The sequence then passes through a stack of 12 transformer blocks, each comprising multi-head self-attention and a feedforward layer, with each followed by a normalization layer. A skip connection is applied between the input and output of the multi-head self-attention module. The [CLS] token obtained from the final transformer block is used as the feature vector $z_i^{vit} \in \mathbb{R}^d$, where $d = 768$. Given a batch of images with size $B$, it is processed by $f_{vit}$ to produce a batch of feature vectors $\{z_i^{vit}\}_{i=1}^B$, which are then processed by the CA module to produce the aggregated feature vector $z_i^T$.

**CA Module.** Our CA module includes a similarity network $f_{sim}$ and an aggregation network $f_{agg}$. For $f_{sim}$, we implement two convolutional layers: the first layer has the same input and output channels, which are 768, while the second layer projects from 768 to 1, *i.e.*, a scalar value for the similarity score. Batch Normalization Ioffe & Szegedy (2015) followed by a LeakyReLU activation is applied between these two convolutional layers. For $f_{agg}$, three convolutional layers are employed, each followed by Batch Normalization and LeakyReLU. The first two layers have the same input and output channels, which are 768, while the last one produces $C$ logits, where $C$ is the number of categories.

### E.2 STUDENT NETWORK

**Feature Extractor.** The ResNet family He et al. (2016) and AlexNet Krizhevsky et al. (2012) are adopted as feature extractors for the student network, *i.e.*, $f_{cnn}$. The ResNet architectures used in this study include ResNet101, ResNet50, ResNet34, and ResNet18. The specific network applied to the student network depends on the dataset and settings to ensure a fair comparison with existing studies or for evaluation purposes. Given an input image $x_i$, the feature extractor $f_{cnn}$ produces a feature vector $z_i^{cnn} \in \mathbb{R}^{d'}$, where $d'$ represents the dimensionality of the feature vector. The value of $d'$ depends on the specific feature extractor used: $d' = 512$ for ResNet18 and ResNet34, $d' = 2048$ for ResNet50 and ResNet101, and $d' = 4096$ for AlexNet.

**MLP Classifier.** The MLP classifier includes two linear layers. The first layer projects from a $d'$ dimension to $512$, and the second layer projects from $512$ to $C$ logits. Between the two linear layers, there is a normalization operation.

### E.3 COMPARISON BETWEEN CA AND MLP

We further highlight the differences between CA and MLP and analyze some significant time complexities. For a batch of $B$ samples, CA constructs a similarity matrix $\tilde{S} \in \mathbb{R}^{B \times B}$, using $f_{sim}$ for scoring and $f_{agg}$ for feature aggregation and logits generation. Constructing the similarity matrix requires computing all possible pairwise feature differences in the batch, with a time complexity of $\mathcal{O}(B^2 \cdot d)$, where $d$ is the feature vector dimension of $f_{vit}$. $f_{sim}$ then assigns scalar similarity scores. Subsequently, feature aggregation involves a weighted sum of each feature vector with all others, costing $\mathcal{O}(B^2 \cdot d)$, followed by concatenation ($\mathcal{O}(B \cdot d)$) and $f_{agg}$ to produce $C$-dimensional logits. In this manner, **the CA module projects aggregated feature vectors into the logit space**. In contrast, **the MLP generates the logits for each feature vector independently using a single network**. Although the proposed CA module introduces higher complexity, it is used only during training, whereas MLP integrated with CNN-based networks is used during inference, ensuring practicality.

## F ADDITIONAL RESULTS

This section presents extensive experimental results on UDA and SSDA settings.

### F.1 UNSUPERVISED DOMAIN ADAPTATION

**VisDA2017.** The classification accuracy for each class is listed in Tab. 8. Using ResNet101 as the backbone, the proposed student network GraDA (**S**) achieves the highest classification accuracy across all classes except for the "*car*" class. The average accuracy over the 12 classes for the student network reaches $96.5\%$, representing a $6.5\%$ improvement over the second-best method, HVCLIP Vesdapunt et al. (2024). Compared to prior works using ViT-B as a backbone, the teacher network GraDA (**T**) also achieves the best accuracy of $97.0\%$. Notably, the mean accuracy gap between the student and teacher networks is marginal, measuring only $0.5\%$.

**Mini-DomainNet.** We reported the classification results of 12 DA tasks in Tab. 9. The average accuracy is $93.1\%$, surpassing the second-best method, ECB Ngo et al. (2024), by $8.9\%$.

**Office-31** & **ImageCLEF-DA.** Tables 10 and 11 present the classification results for **Office-31** and **ImageCLEF-DA**, respectively. The proposed student model, utilizing ResNet50, achieves competitive accuracy on both datasets, with $96.5\%$ on **Office-31** and $94.8\%$ on **ImageCLEF-DA**.

### F.2 SEMI-SUPERVISED DOMAIN ADAPTATION (SSDA)

The proposed method can be easily extended to the SSDA setting, where a limited number of labeled target samples per class are available, $D_{tar}^l = \{(x_i^{tar}, y_i^{tar})\}_{i=1}^{N_{tar}^l}$, where $N_{tar}^l$ is the number of labeled target samples and $N_{tar}^l \ll N_{tar}$. We simply add $D_{tar}^l$ into $D_{cb}$, which can be formed as follows:

$$D_{cb} = D_{src} \cup D_{ps} \cup D_{tar}^l, N_{cb} = N_{src} + N_{ps} + N_{tar}^l.$$

| Net | Method | I→P | P→I | I→C | C→I | C→P | P→C | Mean |
|---|---|---|---|---|---|---|---|---|
| ResNet50 | MCD (CVPR'18) | 77.3 | 89.2 | 92.7 | 88.2 | 71.0 | 92.3 | 85.1 |
| | GVB-GD (CVPR'20) | 78.2 | 92.7 | 96.5 | 91.5 | 78.2 | 95.0 | 88.7 |
| | VRDA (ICASSP'22) | 78.3 | 93.8 | 96.3 | 93.5 | 78.0 | 96.3 | 89.4 |
| | DALN (CVPR'22) | 80.5 | 93.8 | 97.5 | 92.8 | 78.3 | 95.0 | 89.7 |
| | CKB (CVPR'21) | 80.7 | 93.7 | 97.0 | 93.5 | 79.2 | 97.0 | 90.2 |
| | AML (IEEE Trans'23) | 80.8 | 93.8 | 97.7 | 93.2 | 80.2 | 98.2 | 90.7 |
| | GOAL (TPAMI'24) | 82.2 | 94.1 | 97.3 | 95.6 | 82.3 | 96.4 | 91.3 |
| | ☞ GraDA (S) | **85.8** | **99.5** | **99.5** | **99.3** | **84.8** | **99.8** | **94.8** |
| ResNet34 | ☞ GraDA (S) | 85.8 | 99.3 | 99.5 | 99.3 | 84.7 | 99.7 | 94.7 |
| ResNet18 | ☞ GraDA (S) | 84.6 | 99.7 | 99.7 | 99.5 | 84.8 | 99.0 | 94.6 |
| ViT-B | ☞ GraDA (T) | 86.8 | 99.8 | 100.0 | 99.8 | 87.2 | 100.0 | 95.6 |

Table 11: Accuracy (%) on **ImageCLEF-DA** under UDA with various versions of ResNet such as ResNet50, ResNet34, and ResNet18. The best classification accuracy is marked as **bold**.

Then, the training process for SSDA is conducted in the same manner as the unsupervised domain adaptation (UDA).

**Office-Home.** As reported in Tab. 12, we compare our student network with previous SSDA works on the **Office-Home** dataset. Remarkably, in terms of mean accuracy, our GraDA (S) surpasses FMLM Basak & Yin (2024) by 19.4% in the 1-shot setting and EFTL He et al. (2024) by 16.1% in the 3-shot setting. Furthermore, our method is not affected by the addition of labeled target samples, with a gain of only 0.6% from the 1-shot to the 3-shot setting.

**Office-31.** We evaluate our method using the lightweight model, AlexNet Krizhevsky et al. (2012), on **Office-31** under the SSDA setting, as shown in Tab. 13. In the 1-shot and 3-shot settings, our GraDA (S) achieves the new state-of-the-art method with a mean accuracy of 90.8% and 91.7%, respectively.

# G   DISCUSSION

This section further explores the variety of student networks in GraDA and the pivotal role of CA in teacher network design. We also take a closer look at its impact on student performance through the pseudo-label generation process.

## G.1   ROBUSTNESS WITH VARIOUS STUDENT NETWORKS

We investigate the effectiveness of gradient-based knowledge distillation across various student networks. Figures 5 and 6 illustrate results of different versions of the student networks implemented with ResNet50, ResNet34, and ResNet18 for **Office-Home** and ResNet101, ResNet50, and ResNet18 for **VisDA2017**, respectively. Additionally, Tabs. 10 and 11 provide classification accuracy results of various student networks employing ResNet50, ResNet34, and ResNet18 on **Office-31** and **ImageCLEF-DA**. The results presented in these figures and tables demonstrate robustness across various student networks, where smaller networks, such as ResNet18, with relatively fewer parameters, can achieve competitive results compared to larger networks like ResNet50, as shown in Fig. 5, Tabs. 10 and 11, or ResNet101, as illustrated in Fig. 6. For small datasets such as **Office-31** and **ImageCLEF-DA**, the student network based on ResNet18 achieves performance comparable to those based on ResNet50, with only a minimal gap of 0.2%. For moderate and more challenging datasets, such as **Office-Home** and **VisDA2017**, the student network using ResNet18 also demonstrates strong flexibility, closely aligning with the classification results of larger student networks based on ResNet50 or ResNet101.

## G.2   ABILITY OF THE TEACHER NETWORK

Selecting a strong teacher is the most important aspect of knowledge distillation (KD), with various perspectives and definitions. Traditional KD methods often assume that a strong teacher is a large model size. However, (Beyer et al., 2022) argue that a strong teacher is one that is patiently trained

| Net | Method | Ar→Cl | Ar→Pr | Ar→Rw | Cl→Ar | Cl→Pr | Cl→Rw | Pr→Ar | Pr→Cl | Pr→Rw | Rw→Ar | Rw→Cl | Rw→Pr | Mean |
|---|---|---|---|---|---|---|---|---|---|---|---|---|---|---|
| | | | | | | | 1-shot | | | | | | | |
| ResNet34 | DECOTA (ICCV'21) | 42.1 | 68.5 | 72.6 | 60.3 | 70.4 | 70.7 | 60.0 | 48.8 | 76.9 | 71.3 | 56.0 | 79.4 | 64.8 |
| | APE (ECCV'20) | 53.9 | 76.1 | 75.2 | 63.6 | 69.8 | 72.3 | 63.6 | 58.3 | 78.6 | 72.5 | 60.7 | 81.6 | 68.9 |
| | MME (ICCV'19) | 59.6 | 75.5 | 77.8 | 65.7 | 74.5 | 74.8 | 64.7 | 57.4 | 79.2 | 71.2 | 61.9 | 82.8 | 70.4 |
| | CLDA (NIPS'21) | 56.3 | 76.1 | 79.3 | 66.3 | 73.9 | 76.3 | 66.2 | 55.9 | 81.0 | 72.6 | 60.2 | 83.2 | 70.6 |
| | CDAC (CVPR'21) | 61.2 | 75.9 | 78.5 | 64.5 | 75.1 | 75.3 | 64.6 | 59.3 | 80.0 | 72.7 | 61.9 | 83.1 | 71.0 |
| | SPA (NIPS'23) | 62.3 | 76.7 | 79.0 | 66.6 | 77.3 | 76.4 | 65.7 | 59.1 | 80.7 | 71.4 | 65.2 | 84.1 | 72.0 |
| | SLA (CVPR'23) | 63.0 | 78.0 | 79.2 | 66.9 | 77.6 | 77.0 | 67.3 | 61.8 | 80.5 | 72.7 | 66.1 | 84.6 | 72.9 |
| | MCL (IJCAI'22) | 64.4 | 79.5 | 81.2 | 68.5 | 79.3 | 78.4 | 68.0 | 61.1 | 81.3 | 73.8 | 67.0 | 85.5 | 74.0 |
| | EFTL (AAAI'24) | 65.7 | 80.5 | 80.8 | 65.6 | 79.6 | 77.5 | 68.7 | 63.3 | 82.6 | 74.3 | 66.6 | 87.2 | 74.4 |
| | ProML (IJCAI'23) | 64.5 | 79.7 | 81.7 | 69.1 | 80.5 | 79.0 | 69.3 | 61.4 | 81.9 | 73.7 | 67.5 | 86.1 | 74.6 |
| | FMLM (ECCV'24) | 64.1 | 80.1 | 81.1 | 70.6 | 79.5 | 79.1 | 67.9 | 62.5 | 80.9 | 75.2 | 69.1 | 87.9 | 74.8 |
| | ☞ GraDA (S) | **89.9** | **95.5** | **95.8** | **93.5** | **96.3** | **96.9** | **93.7** | **89.1** | **96.9** | **94.5** | **91.2** | **96.9** | **94.2** |
| ViT-B | ☞ GraDA (T) | 90.4 | 95.6 | 96.0 | 93.8 | 96.4 | 97.1 | 93.8 | 89.6 | 97.1 | 94.6 | 91.5 | 97.0 | 94.4 |
| | | | | | | | 3-shot | | | | | | | |
| ResNet34 | MME (ICCV'19) | 63.6 | 79.0 | 79.7 | 67.2 | 79.3 | 76.6 | 65.5 | 64.6 | 80.1 | 71.3 | 64.6 | 85.5 | 73.1 |
| | APE (ECCV'20) | 63.9 | 81.1 | 80.2 | 66.6 | 79.9 | 76.8 | 66.1 | 65.2 | 82.0 | 73.4 | 66.4 | 86.2 | 74.0 |
| | CDAC (CVPR'21) | 65.9 | 80.3 | 80.6 | 67.4 | 81.4 | 80.2 | 67.5 | 67.0 | 81.9 | 72.2 | 67.8 | 85.6 | 74.2 |
| | SPA (NIPS'23) | 63.1 | 81.0 | 80.2 | 68.5 | 81.7 | 77.5 | 69.5 | 65.2 | 82.0 | 73.9 | 67.2 | 87.0 | 74.7 |
| | CLDA (NIPS'21) | 63.4 | 81.4 | 81.3 | 70.5 | 80.9 | 80.3 | 72.4 | 63.9 | 82.2 | 76.7 | 66.0 | 87.6 | 75.5 |
| | DECOTA (ICCV'21) | 64.0 | 81.8 | 80.5 | 68.0 | 83.2 | 79.0 | 69.9 | 68.0 | 82.1 | 74.0 | 70.4 | 87.7 | 75.7 |
| | SLA (CVPR'23) | 67.3 | 82.6 | 81.4 | 69.2 | 82.1 | 80.1 | 70.1 | 69.3 | 82.5 | 73.9 | 70.1 | 87.1 | 76.3 |
| | MCL (IJCAI'22) | 67.5 | 83.9 | 82.4 | 71.4 | 84.3 | 81.6 | 69.9 | 68.0 | 83.0 | 75.3 | 70.1 | 88.1 | 77.1 |
| | ProML (IJCAI'23) | 67.8 | 83.9 | 82.2 | 72.1 | 84.1 | 82.3 | 72.5 | 68.9 | 83.8 | 75.8 | 71.0 | 88.6 | 77.8 |
| | FMLM (ECCV'24) | 68.8 | 84.7 | 84.2 | 70.6 | 83.7 | 82.4 | 70.5 | 70.9 | 84.3 | 75.7 | 71.1 | 88.5 | 77.9 |
| | EFTL (AAAI'24) | 70.3 | 84.8 | 83.8 | 70.6 | 84.6 | 81.5 | 72.6 | 70.9 | 85.4 | 77.5 | 72.8 | 89.3 | 78.7 |
| | ☞ GraDA (S) | **91.0** | **96.1** | **97.3** | **93.8** | **96.3** | **97.5** | **94.1** | **90.7** | **97.1** | **94.5** | **92.6** | **97.0** | **94.8** |
| ViT-B | ☞ GraDA (T) | 91.7 | 96.1 | 97.4 | 94.1 | 96.5 | 97.8 | 94.3 | 91.0 | 97.3 | 94.7 | 93.1 | 97.1 | 95.1 |

Table 12: Accuracy (%) on **Office-Home** under the SSDA setting. The best classification accuracy is marked as **bold**.

| Net | Method | W→A | | D→A | | Mean | |
|---|---|---|---|---|---|---|---|
| | | 1-shot | 3-shot | 1-shot | 3-shot | 1-shot | 3-shot |
| AlexNet | MME (ICCV'19) | 57.2 | 67.3 | 55.8 | 67.8 | 56.5 | 67.6 |
| | BiAT (IJCAI'20) | 57.9 | 68.2 | 54.6 | 68.5 | 56.3 | 68.4 |
| | CDAC (CVPR'21) | 63.4 | 70.1 | 62.8 | 70.0 | 63.1 | 70.0 |
| | CLDA (NIPS'21) | 64.6 | 70.5 | 62.7 | 72.5 | 63.6 | 71.5 |
| | ECB (CVPR'24) | 77.9 | 85.2 | 76.3 | 84.0 | 77.1 | 84.6 |
| | ☞ GraDA (S) | **91.1** | **92.1** | **90.4** | **91.3** | **90.8** | **91.7** |
| ViT-B | ☞ GraDA (T) | 91.9 | 92.8 | 91.2 | 92.4 | 91.6 | 92.6 |

Table 13: Accuracy (%) on **Office-31** under the SSDA setting. The best classification accuracy is marked as **bold**.

over an extended period, producing consistent and reliable outputs. (Martin et al., 2023) propose that a strong teacher dynamically adjusts the amount of knowledge transfer based on the feature gap between the teacher and student models. Similarly, (Sengupta et al., 2024) define a strong teacher as one that can both collaborate with and compete against the student network during the distillation process. However, these approaches typically employ a combination of a feature extractor and an MLP classifier. This setup focuses on *processing individual inputs without considering their neighboring information*. Consequently, the previous teacher networks had limited ability to construct and generalize extracted knowledge effectively.

This section provides further insights into the capabilities of our teacher network. As discussed in the *main manuscript*, the CA module plays a pivotal role in the success of the teacher network's architecture. It effectively explores intra-class relations within each domain, thereby enriching category representations. Additionally, it facilitates class-aware feature alignment across domains, addressing the domain shift issue. To illustrate the superior effectiveness of the CA module, we present additional visualizations of the similarity matrix $\tilde{S}$ on **Office-Home** under the UDA setting and on **DomainNet** under the 3-shot SSDA setting. We use the visualization results from the *left-side figures* in Tabs. 16 and 17 to analyze **the insight operation of our teacher network during training**, including:

| Teacher-Student Pair | #Params (M) | Ar→Cl | Ar→Pr | Ar→Rw | Cl→Ar | Cl→Pr | Cl→Rw | Pr→Ar | Pr→Cl | Pr→Rw | Rw→Ar | Rw→Cl | Rw→Pr | Mean |
|---|---|---|---|---|---|---|---|---|---|---|---|---|---|---|
| ViT-B+CA (T) | 89.7 | 89.3 | 94.8 | 97.2 | 94.1 | 93.8 | 96.2 | 92.8 | 89.1 | 97.4 | 95.1 | 91.5 | 97.7 | 94.0 |
| ResNet50+MLP (S) | 24.6 | 88.6 | 94.8 | 97.0 | 93.9 | 93.7 | 96.0 | 92.7 | 88.3 | 97.2 | 95.0 | 90.9 | 97.6 | 93.8 |
| ViT-tiny+MLP (S) | 5.7 | 87.0 | 93.1 | 96.2 | 93.6 | 93.8 | 95.7 | 92.1 | 87.0 | 96.0 | 94.6 | 88.2 | 96.9 | 92.9 |
| ResNet50+CA (T) | 30.3 | 78.3 | 87.6 | 95.3 | 81.5 | 83.9 | 89.7 | 84.8 | 75.9 | 93.8 | 92.5 | 82.8 | 93.0 | 86.6 |
| ResNet50+MLP (S) | 24.6 | 77.8 | 87.3 | 95.1 | 81.4 | 83.8 | 89.5 | 84.6 | 75.5 | 93.5 | 92.3 | 82.5 | 92.9 | 86.4 |
| ViT-tiny+CA (T) | 8.1 | 68.7 | 84.6 | 89.9 | 82.0 | 81.4 | 84.5 | 83.7 | 69.6 | 93.0 | 87.5 | 75.6 | 91.9 | 82.7 |
| ViT-tiny+MLP (S) | 5.7 | 68.3 | 82.9 | 89.0 | 81.8 | 80.5 | 82.9 | 83.6 | 69.1 | 92.8 | 87.3 | 75.1 | 90.8 | 82.0 |
| HVCLIP (ResNet50) | ≈101.5 | 62.0 | 85.8 | 86.2 | 77.8 | 84.3 | 86.8 | 80.7 | 66.5 | 87.8 | 80.3 | 64.9 | 90.4 | 79.5 |

Table 14: Comparison of teacher-student pairs on **Office-Home** under the UDA setting (*Full version*).

- **Enhancing the generalization by enriching intra-class relations.** We process a batch of $B = 16$ images through the teacher network, covering two classes, as an example. For each class, there are *four samples from the source domain* and another *four samples from the target domain*. For easier visualization, source samples are marked in green, while target samples are marked in pink. The CA module works effectively to show high similarity scores for same-category samples within the source domain indicated by dashed green. This is intuitive, as the source ground truth information supports these results. For target samples, the CA module relies on the quality of generated pseudo labels. Despite this, it still assigns high similarity scores to samples belonging to the same class, as marked by dashed pink boxes.

- **Handling the domain shift via class-aware feature alignment.** To demonstrate the effectiveness of the CA module in the teacher network for handling the domain shift issue between the source and target domains, we present the similarity scores of cross-domain samples highlighted by dashed yellow boxes. The CA module successfully identifies samples belonging to the same class but from different domains by assigning high similarity scores. In contrast, it assigns low similarity scores to samples from distinct classes, ensuring robust class-aware feature alignment.

These observations strongly emphasize the capability of the CA module in teacher network design. Furthermore, unlike previous teacher networks that make predictions directly from features extracted by the feature extractor, our teacher network bases its predictions on aggregated features.

**Effectiveness of the teacher network in the testing phase.** We use the visualization results from the *right-side figures* in Tabs. 16 and 17 to evaluate the effectiveness of the teacher network during testing. As illustrated in these figures, the CA module effectively identifies target samples within the same category by assigning high similarity scores, while providing low similarity scores to samples from distinct classes. This observation highlights the teacher network's capability to exploit intra-class relations within the target data, grouping samples of the same category while ensuring that samples from different classes remain distinguishable.

### G.3 ROLES OF CA MODULE IN PSEUDO-LABELING

To demonstrate the effectiveness of our CA module in the pseudo-labeling process, we compare the quantity and quality of pseudo labels generated by ViT-B+MLP (**Setting 1**) and ViT-B+CA (**Setting 2**) as teacher networks on **VisDA2017**. As shown in Fig. 7, the dashed lines indicate the number of ground-truth labels for each class, while the colored bars represent the pseudo-label counts. These two settings follow the same training manner combining *supervised*, *self-enhanced*, and *cross-class confusion* strategies.

In **Setting 1**, we employ MLP as the classifier, which is unable to capture relationships among neighboring samples. As illustrated in Fig. 7a, the ViT-B+MLP network assigns a higher number of pseudo labels than ground-truth labels to classes such as "*bicycle*", "*bus*", "*skate*", and "*truck*". Obviously, those excessive pseudo labels are incorrect, indicating that ViT-B+MLP does not ensure the quality of its generated pseudo labels, which misguide the student network. Furthermore, as it focuses only on individual representations, the MLP classifier also demonstrates its weakness in differentiating between highly similar classes, such as "*bus*", "*car*", "*train*", and "*truck*".

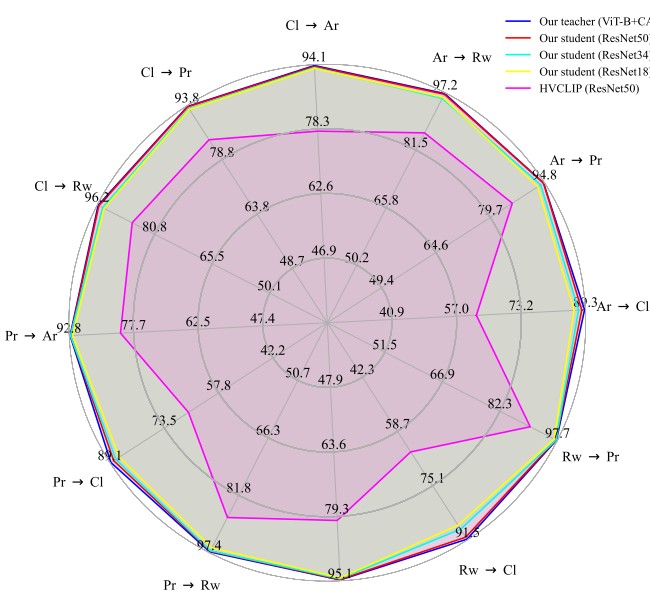

Figure 5: Performance of the teacher network and its three students on the **Office-Home** dataset under the UDA setting. The black numbers on the radar chart indicate the distance values from the center to the corresponding intersections of the concentric circles. The classification results show that our method performs effectively across various student network configurations. It remains unaffected by the capability gap between teacher and student networks and significantly surpasses the second-best approach, HVCLIP (Vesdapunt et al., 2024), even with a student network using fewer parameters.

| Teacher-Student Pair | | #Param. (**M**) | Ar→Cl | Cl→Pr | Pr→Rw | Rw→Ar | Mean |
|---|---|---|---|---|---|---|---|
| **T** | ViT-B+CA | 89.7 | 89.3 | 93.8 | 97.4 | 95.1 | 93.9 |
| **S** | ResNet50+MLP | 24.6 | 88.6 | 93.7 | 97.2 | 95.0 | 93.6 |
| | ResNet34+MLP | 21.6 | 87.8 | 93.2 | 97.1 | 94.8 | 93.2 |
| | ResNet18+MLP | 11.5 | 86.7 | 93.1 | 96.3 | 94.6 | 92.7 |

Table 15: Accuracy of different student network (**S**) paired with the fixed teacher network (**T**) on **Office-Home** under the UDA setting.

In contrast, in **Setting 2**, the proposed CA module enriches intra-class relationships by aggregating representations of samples within the same category. This enables the ViT-B+CA network to enhance the robustness of representations within each class. As a result, the ViT-B+CA network demonstrates stronger discriminability between different classes compared to the ViT-B+MLP network. This can be observed clearly in the *car* class, where the quantity and quality of pseudo labels generated by **Setting 2** are significantly higher than those of **Setting 1**, as shown in Figs. 7a and 7b. By doing so, the ViT-B+CA teacher network can provide more reliable information to train the student network, resulting in a substantial increase in the overall results of the student network.

### G.4 EFFECTIVENESS OF GRADIENT-BASED KD

We first provide the method to visualize the gradient trajectories of teacher and student networks, learnable parameters and their corresponding loss values from all episodes are used: $\{\theta_T^e, l_T^e\}_{e=1}^E$ and $\{\theta_S^e, l_S^e\}_{e=1}^E$, where $\theta_T = \{\theta_{vit}, \theta_{sim}, \theta_{agg}\}$, and $\theta_S = \{\theta_{cnn}, \theta_{mlp}\}$. Here, $l_T^e$ and $l_S^e$ are per-epoch loss values of the teacher and student networks, respectively. The parameters $\{\theta_T^e\}_{e=1}^E$ and

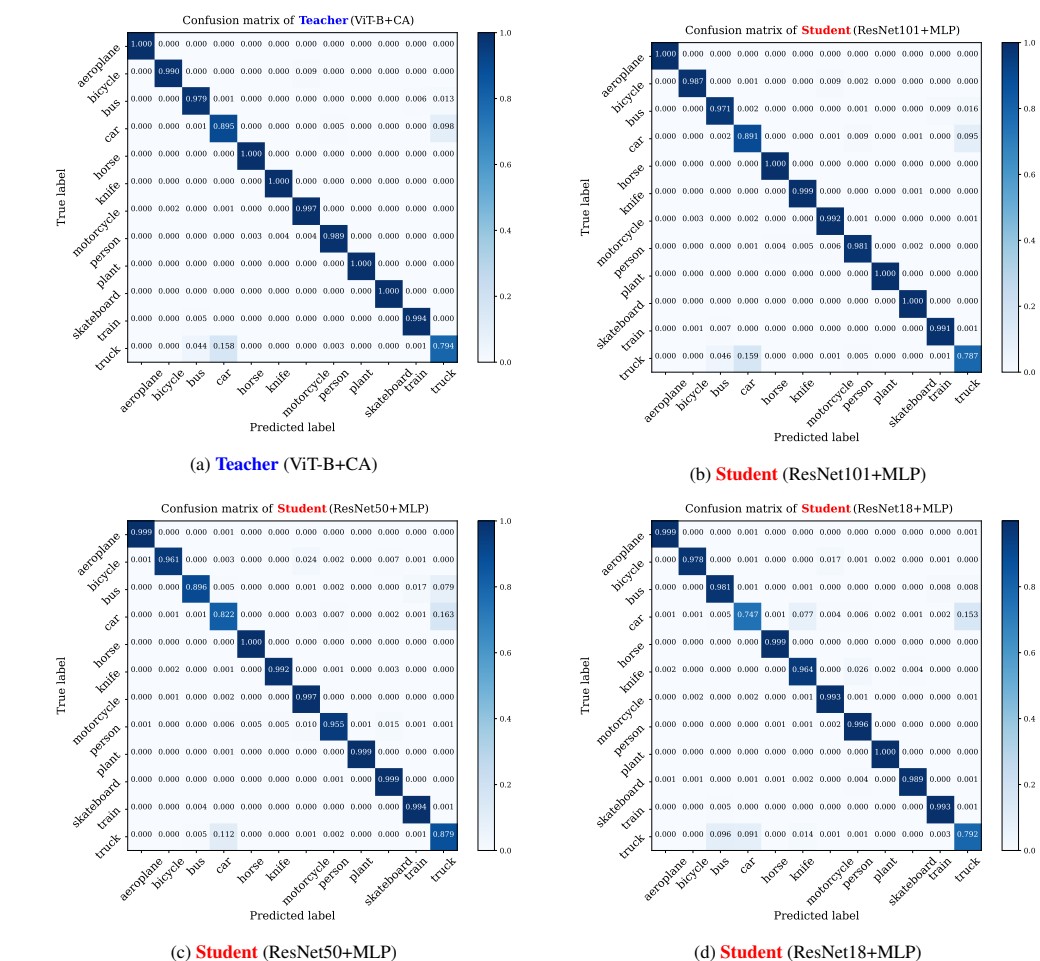

(a) **Teacher** (ViT-B+CA)

(b) **Student** (ResNet101+MLP)

(c) **Student** (ResNet50+MLP)

(d) **Student** (ResNet18+MLP)

Figure 6: Confusion matrix of (a) the teacher network, and (b), (c), (d) representing various student networks, specifically ResNet101, ResNet50, and ResNet18, respectively. These networks are evaluated across 12 classes on **VisDA2017** under the UDA setting.

$\{\theta_S^e\}_{e=1}^E$ are projected into 2D space using UMAP McInnes et al. (2018), yielding $\tilde{\boldsymbol{\theta}}_T = \{\tilde{\theta}_T^e\}_{e=1}^E$ and $\tilde{\boldsymbol{\theta}}_S = \{\tilde{\theta}_S^e\}_{e=1}^E$, where $\tilde{\theta}_T^e, \tilde{\theta}_S^e \in \mathbb{R}^2$. The student projections $\tilde{\boldsymbol{\theta}}_S$ are scaled to match the value range of $\tilde{\boldsymbol{\theta}}_T$. We then sum up the loss values of the two networks per epoch to obtain a set of combined loss values, $L = \{l_T^e + l_S^e\}_{e=1}^E$. The variables $\tilde{\boldsymbol{\theta}}_S$, $\tilde{\boldsymbol{\theta}}_T$, and $L$ are used to estimate the loss landscape using cubic interpolation. Finally, $\tilde{\boldsymbol{\theta}}_S$, $\tilde{\boldsymbol{\theta}}_T$, and the loss landscape are plotted to illustrate the gradient trajectories of the two networks, showing the direction of their convergency in the respective minima.

We provide additional results of the gradient trajectory visualization with various student networks such as ResNet50, ResNet34, and ResNet18 for the Ar→Cl, Ar→Pr, Pr→Ar, and Rw→Cl tasks from the **Office-Home** dataset in Tab. 18. These results demonstrate that the teacher network effectively guides the student, regardless of various student network architectures, within our GraDA framework, further validating its reliability for DA tasks.

### G.5   FAIRNESS OF THE TEACHER NETWORK (*Extended Version*).

This study presents an enhanced approach to fairness evaluation, extending comparisons across multiple teacher networks. Experimental outcomes full 12 domain adaptation tasks, utilizing the **Office-Home** dataset, are detailed in Table 14. Notably, networks employing the robust ViT-B+CA teacher model (denoted as (**S**)) demonstrate superior performance. Furthermore, the proposed CA

module proves effective when integrated with either the ResNet50 architecture or the more compact ViT-Tiny model. The student network (**S**) supported by the CA-based teacher network (**T**) consistently surpasses the second-best method, HVCLIP, highlighting that its performance improvements derive primarily from the CA module and the efficacy of pseudo-labeling, rather than solely from the backbone architecture.

### G.6 CAN THE TEACHER ADAPT TO VARIOUS STUDENTS?

To explore this concern, we construct diverse student networks, including ResNet50+MLP, ResNet34+MLP, and ResNet18+MLP, while fixing ViT-B+CA as the teacher network. As listed in Table 15, the experimental results across 4 DA tasks on **Office-Home** under the UDA setting demonstrate that GraDA is effective regardless of the student network used. Surprisingly, despite having considerably fewer parameters than ResNet34 and ResNet50, the student network based on ResNet18 achieves competitive performance, with small performance gaps of 0.5% and 0.9%, respectively.

## H VISUALIZATION ANALYSIS

In this section, t-SNE Van der Maaten & Hinton (2008) visualizations are provided to show embedding improvements across training strategies, while Grad-CAM Selvaraju et al. (2017) is used to demonstrate the enhanced visual performance of the student network in GraDA.

### H.1 T-SNE VISUALIZATION

We use t-SNE Van der Maaten & Hinton (2008) to further evaluate the effectiveness of the proposed gradient-based knowledge distillation for domain adaptation via visualizing domain alignment and target feature distributions. Figure 8 presents the visualization results of the Rw→Cl task on **Office-Home** under the UDA setting, while Fig. 9 illustrates results of the rel→pnt task on **DomainNet** under the SSDA setting (3-shot). We first show results of the vanilla student trained using *supervised*, *self-enhanced*, and *cross-class confusion* losses without guidance from the teacher network, as specified in setting **S3** of the *main manuscript*. Then, we investigate the impact of the teacher network on the feature space of the student network by progressively adding *supervised* (**Teacher**+**S4**), *self-enhanced* (**Teacher**+**S5**), and *cross-class confusion* (**Teacher**+**S6**) settings.

The vanilla student (**S3**) struggles to provide robust representations due to the sensitivity of CNN to domain shift and its limited ability to capture relationships among neighboring samples. As shown in Figs. 8a and 9a, the target features are highly misalignment compared to those guided by the teacher network under the **Teacher**+**S4** setting (Figs. 8b and 9b) thanks to ability in enriching intra-class relations. In cases **Teacher**+**S5**, the teacher network leverages pseudo labels generated from unlabeled target data to enhance intra-class information on the target domain and mitigating domain shifts through *class-aware feature alignment*. As shown in Figs. 8c and 9c, the teacher network effectively guides the student network using these pseudo labels, enabling the student to align source and target features. Additionally, the discriminative ability among the different classes of the student network is also improved, as illustrated in Figs. 8g and 9g. Finally, we implement the cross-class confusion loss in the **Teacher**+**S6** setting to reduce ambiguous prediction among classes, resulting in a slight improvement, as shown in Figs. 8d and 8h for **Office-Home**, and Figs. 9d and 9h for **DomainNet**.

### H.2 GRAD-CAM VISUALIZATION

We visualize attention maps to examine our gradient-guided ability of the teacher network for the student network by using Grad-CAM Selvaraju et al. (2017). To clearly demonstrate the improvements of our student network, GraDA (**S**), over the vanilla student network (without any guidance), we present samples that are misclassified by the vanilla student but correctly classified by GraDA (**S**) under the UDA and SSDA settings on the **Office-Home** and **DomainNet** datasets.

**Office-Home.** For the **Office-Home** dataset under UDA, we present attention results for four tasks: Ar→Cl, Cl→Pr, Pr→Rw, and Rw→Ar. In both the vanilla student network and GraDA (**S**), ResNet50 is utilized as the feature extractor. As shown in Tab. 19, without any guidance, the

vanilla student network struggles to capture object regions in several classes, such as "*alarm clock*", "*keyboard*", "*spoon*", "*bucket*", *chair*", "*couch*", "*flipflops*", "*flowers*", and "*shelf*". Meanwhile, GraDA (**S**) is shown to focus more precise regions, shifting focus from irrelevant to relevant elements, such as from a cable to a "*keyboard*" or a broom to a "*bucket*". Upon closer examination, it is clear that GraDA (**S**) closely emulates the behavior of GraDA (**T**), which itself demonstrates strong performance. For example, in the case of the class "*bed*", the vanilla student network fails to capture the entire bed and instead focuses only on the footboard, leading to the wrong prediction. However, our teacher network successfully captures the full object, enabling our student network, GraDA (**S**), to learn and mimic this behavior. Similar observations are evident for "*batteries*", "*post-it notes*", "*toothbrush*", and "*toys*".

**DomainNet.** For the **DomainNet** dataset under SSDA, we extract the attention results for four tasks: rel→clp, clp→skt, skt→pnt, and pnt→rel, which are shown in Tab. 20. ResNet34 is utilized as a feature extractor for the student network. Overall, we observe that the teacher network GraDA (**T**) accurately captures the salient regions that strongly represent the class label of the image. For samples containing a single instance, such as "*crab*," the teacher network accurately focuses on the crab in the center, effectively guiding the student to mimic this behavior. In contrast, the vanilla student focuses only on the frame, which provides no key information and ultimately leads to misclassification. Same observations are evident for "*cell phone*", "*spider*", "*alarm clock*", "*bus*", and "*submarine*". In the case of multiple instances appearing in an image, two scenarios can be identified: 1) instances with similar characteristics and 2) salient instances that are mixed with miscellaneous or less relevant instances. In the first scenario, for example, an ant appears with a book in the "*ant*" class, and a cello appears with a panda in the "*cello*" class. This can lead to confusion. Interestingly, the student network guided by GraDA (**T**) is shown to correctly focus on the salient instances, precisely detecting the ant's head in the "*ant*" class and effectively separating the cello from the panda in the "*cello*" class. In contrast, the vanilla student fails to do so, focusing on completely irrelevant instances in all cases. Similar observations are evident for the "*cactus*", "*cell phone*", and "*banana*" classes. In the second scenario, where multiple instances with similar characteristics appear, our student network successfully covers all of them. For example, GraDA (**S**) captures all dolphins and all rabbits in the "*dolphin*" and "*rabbit*" classes, respectively, whereas the vanilla student focuses on only one instance. Similar observations are made for the "*whale*" and "*sheep*" classes.

| Intra-class relations and Class-aware alignment (during training) | Intra-class relations on target domain only (during testing) |
| --- | --- |

*Art→Clipart*

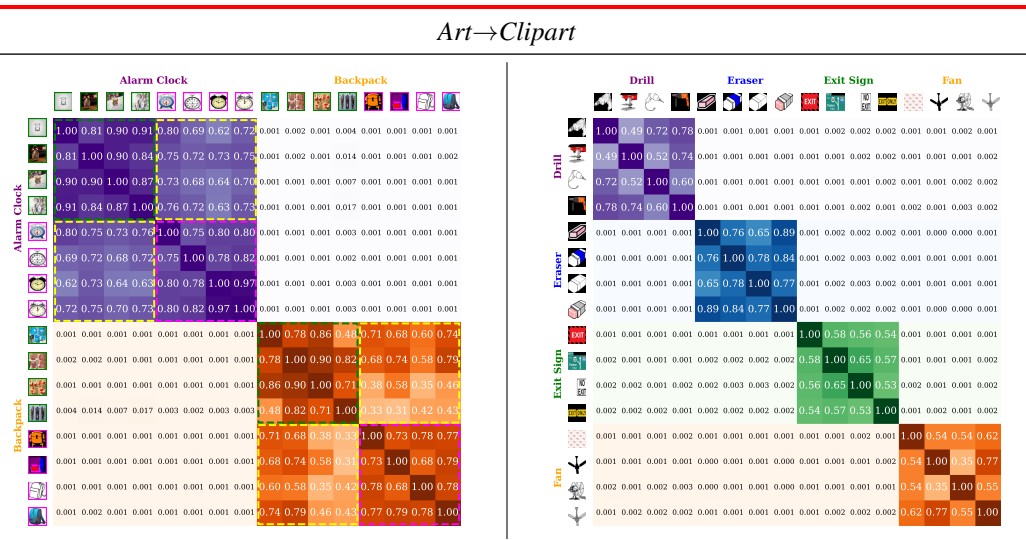

| Intra-class relations and Class-aware alignment (during training) | Intra-class relations on target domain only (during testing) |
|---|---|

*Art→Product*

*Art→Real World*

*Clipart→Art*

| Intra-class relations and Class-aware alignment (during training) | Intra-class relations on target domain only (during testing) |
|---|---|

*Clipart→Product*

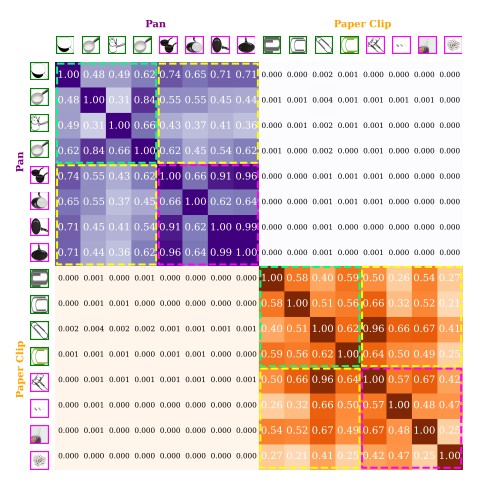 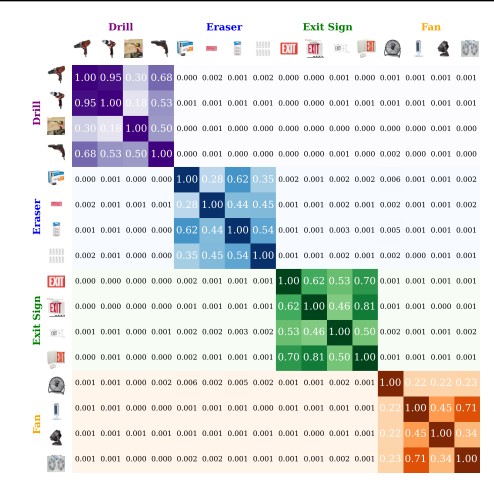

*Clipart→Real World*

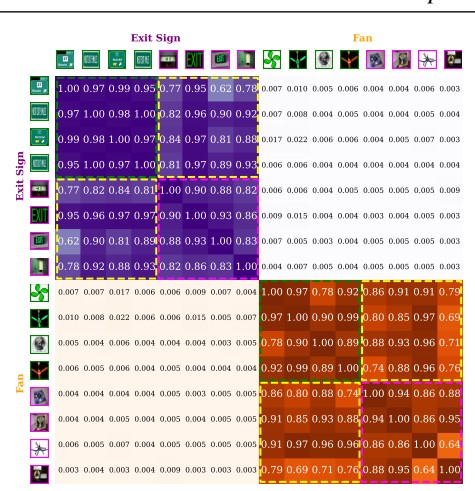 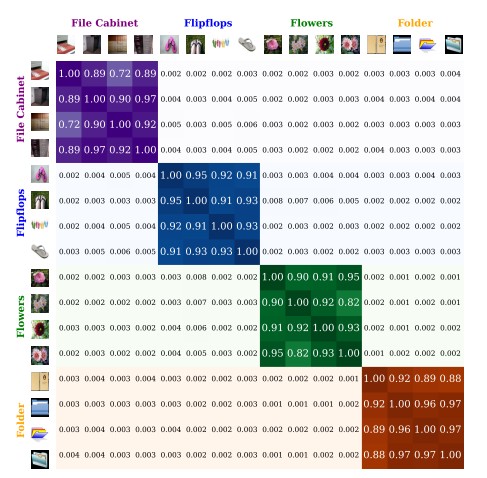

*Product→Art*

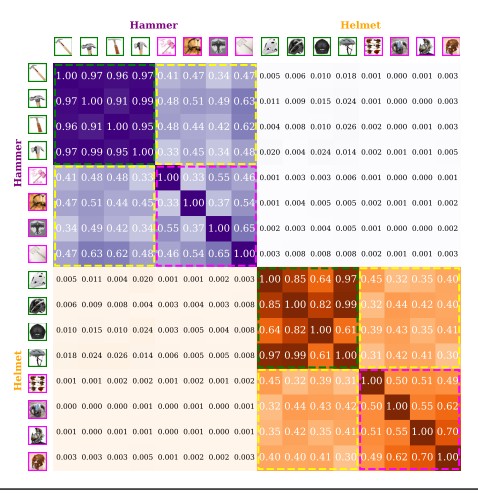 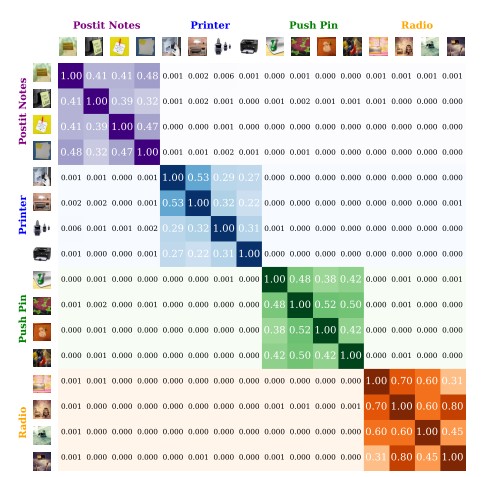

## Intra-class relations and Class-aware alignment (during training)

## Intra-class relations on target domain only (during testing)

*Product→Clipart*

*Product→Real World*

*Real World→Art*

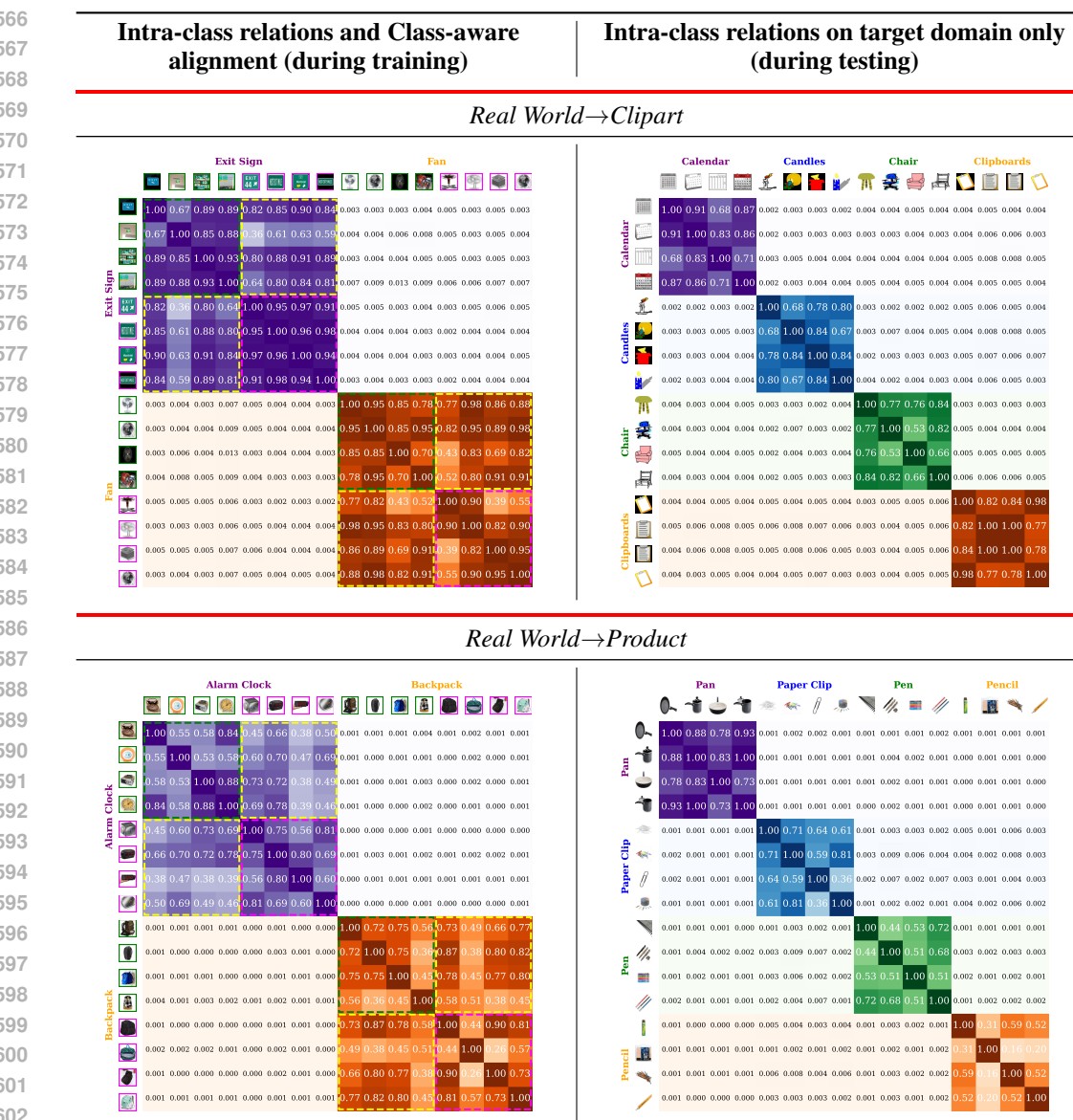

Table 16: Similarity matrix $\tilde{S}$ of the 12 UDA tasks on the **Office-Home** dataset. Dashed green and pink boxes are marked for the relationships of samples within the source and target domains, respectively. The dashed yellow boxes outline the relationships of cross-domain samples. Higher values reflect greater similarity scores.

| Intra-class relations and Class-aware alignment (during training) | Intra-class relations on target domain only (during testing) |
| --- | --- |

*real→clipart*

*real→painting*

*painting→clipart*

| **Intra-class relations and Class-aware alignment (during training)** | **Intra-class relations on target domain only (during testing)** |
| --- | --- |

*clipart→sketch*

*sketch→painting*

*real→sketch*

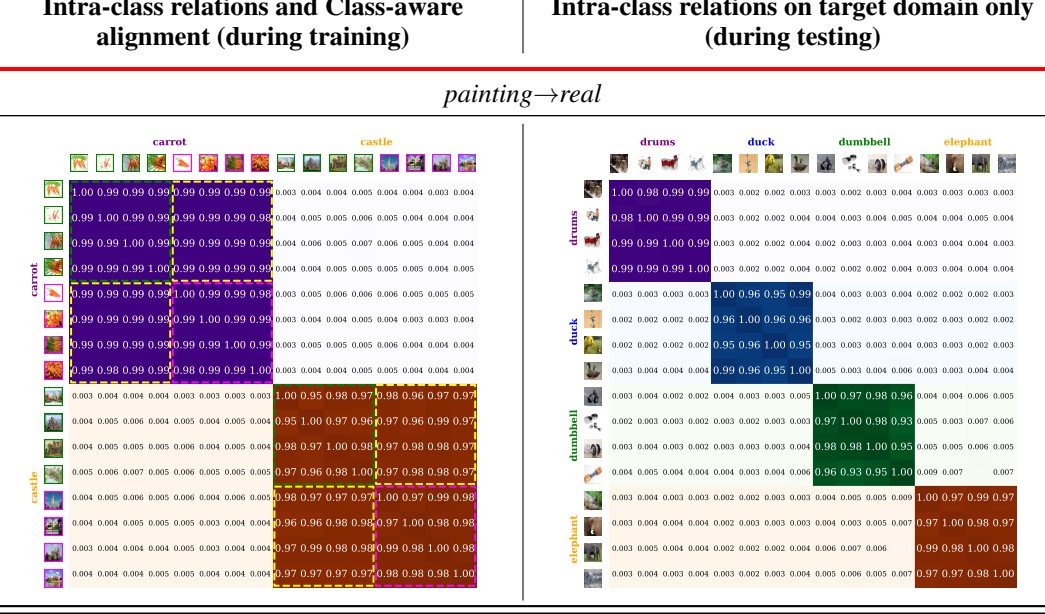

Table 17: The similarity matrix $\tilde{S}$ of the 7 SSDA tasks on the **DomainNet** dataset.

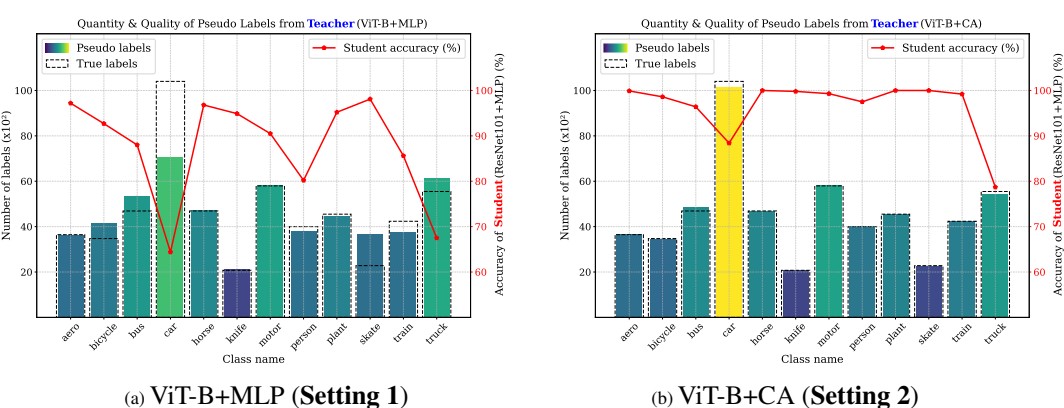

(a) ViT-B+MLP (**Setting 1**)  (b) ViT-B+CA (**Setting 2**)

Figure 7: Comparison of the quantity and quality of pseudo labels between two different teachers (a) ViT-B+MLP and (b) ViT-B+CA on **VisDA2017** under the UDA setting. The bar plots illustrate the number of true labels (outlined in dash lines) and pseudo labels (filled with color) across 12 classes. The red line represents the classification accuracy of the student network (ResNet101+MLP) for each class.

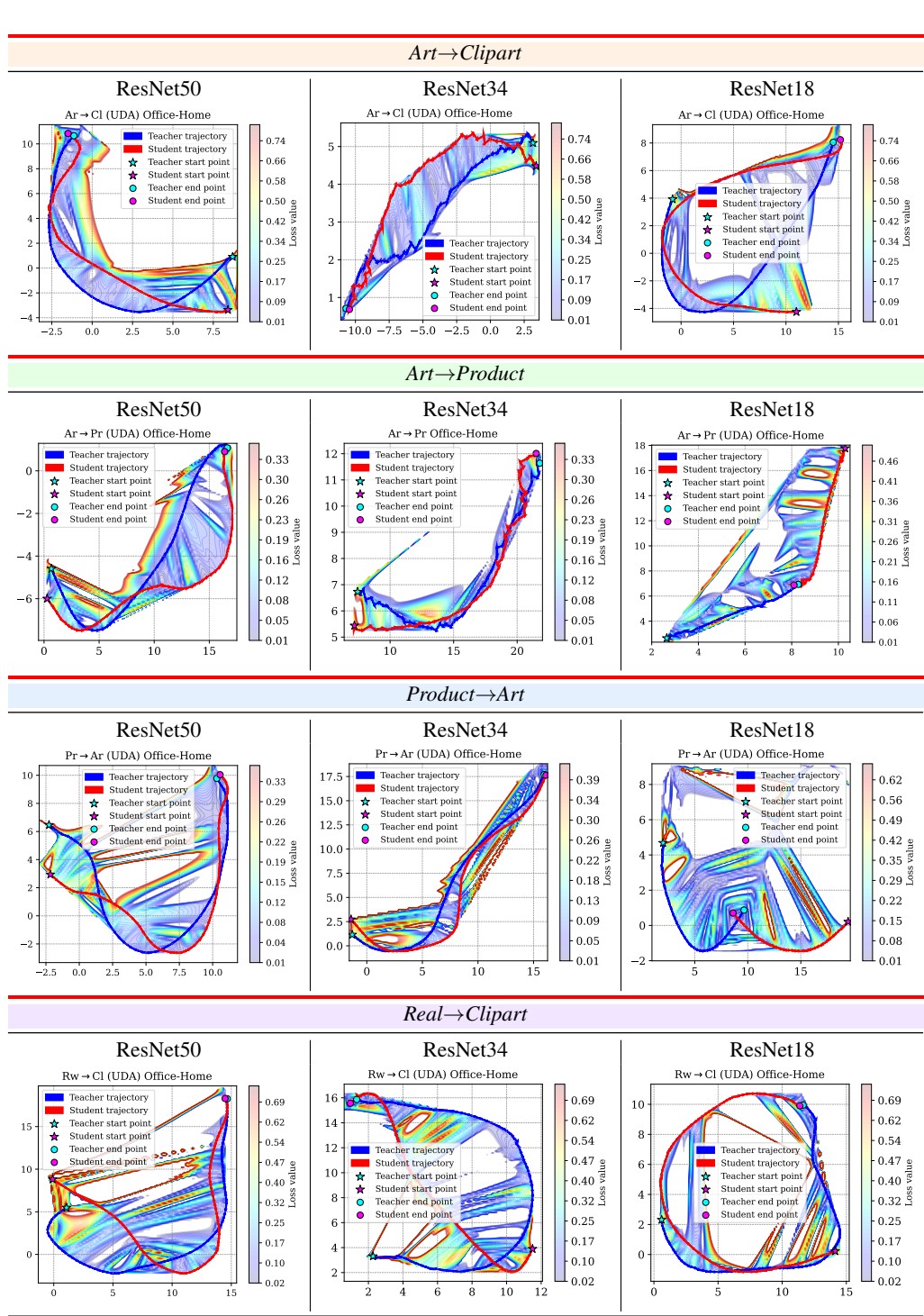

Table 18: 2D visualization of the convergence trajectory in the loss landscape of the teacher network with various student networks.

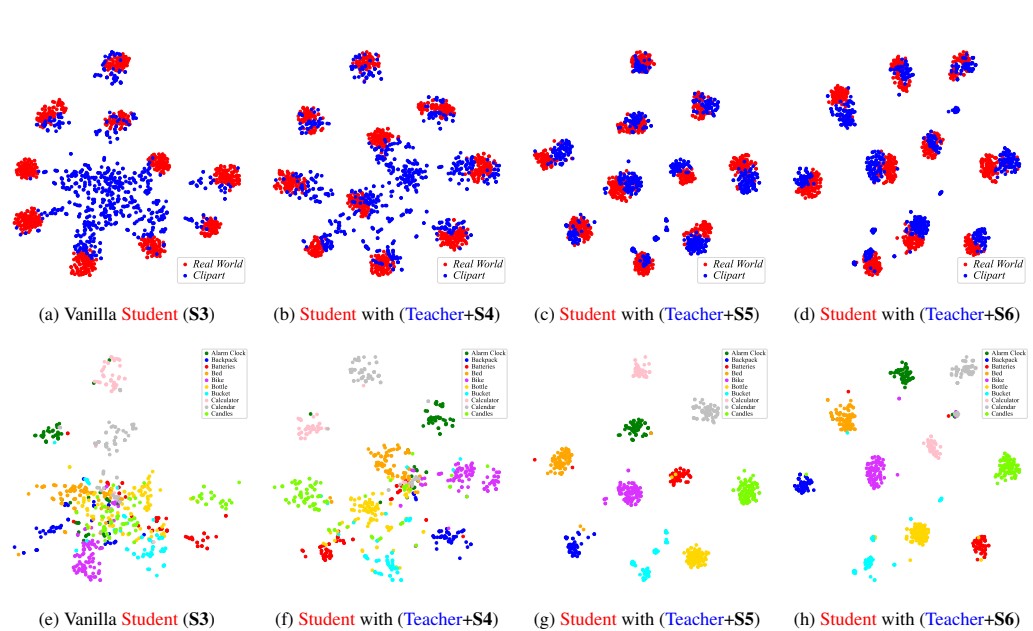

Figure 8: Feature visualization of the student network under different settings. We use t-SNE to visualize for 10 classes of the *Real World→Clipart* task on **Office-Home** under the UDA setting. In (a) and (e), the student network is trained by setting **S3** without the support of teacher guidance. In (b), (c), (d), (f), (g), and (h), the student network is guided by the teacher network, progressively adding **S4** (Supervised), **S5** (Self-Enhanced), and **S6** $\left(\mathcal{L}_{cc}^{T}(p_T(x_i^{tar}))\right)$, respectively. For easy identification of domain alignment features, source features are represented by red markers, and target features by blue markers in Figs. (a), (b), (c), and (d). Target features are shown in Figs. (e), (f), (g), and (h), we use 10 distinct colors to indicate the 10 classes.

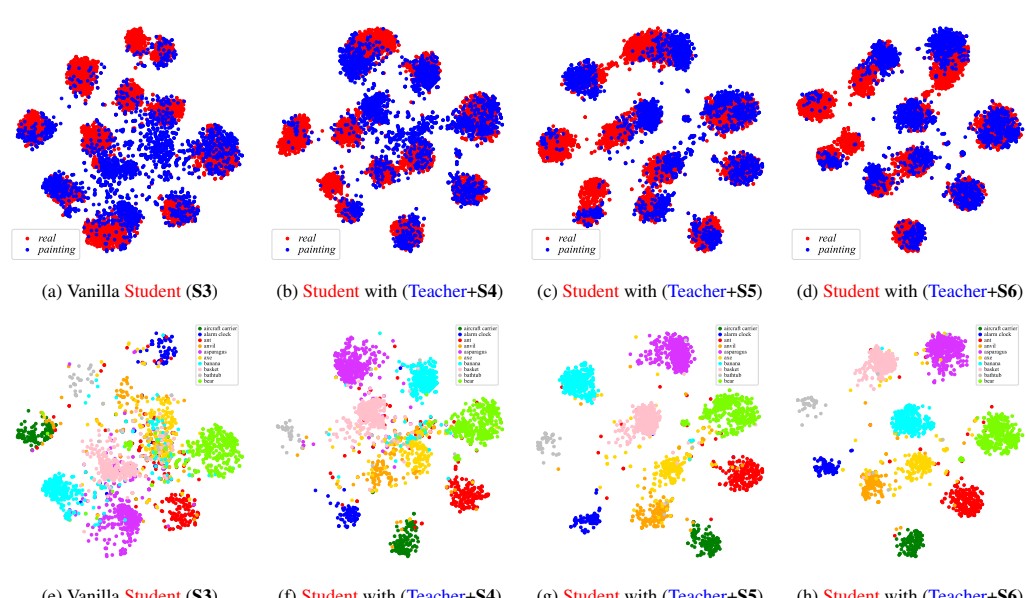

Figure 9: Feature visualization of the student network under different settings on **DomainNet** (3-shot SSDA, *real→painting*).

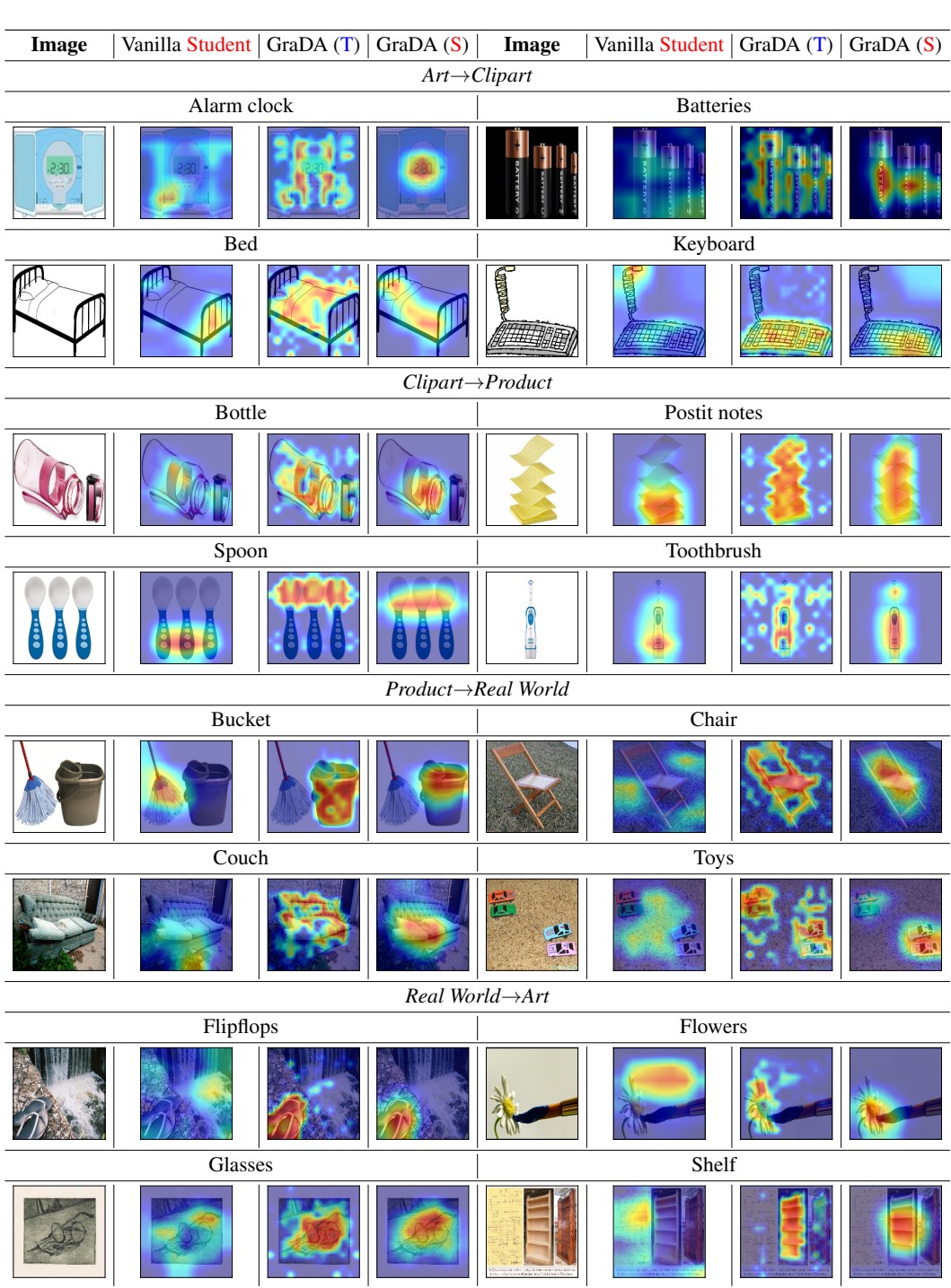

Table 19: Attention maps of the teacher network GraDA (T), and the student network with the vanilla and GraDA (S) variants, on **Office-Home** under the UDA setting. We use Grad-CAM Selvaraju et al. (2017) to identify class-discriminative regions in 4 various samples for each task: Ar→Cl, Cl→Pr, Pr→Rw, and Rw→Ar.

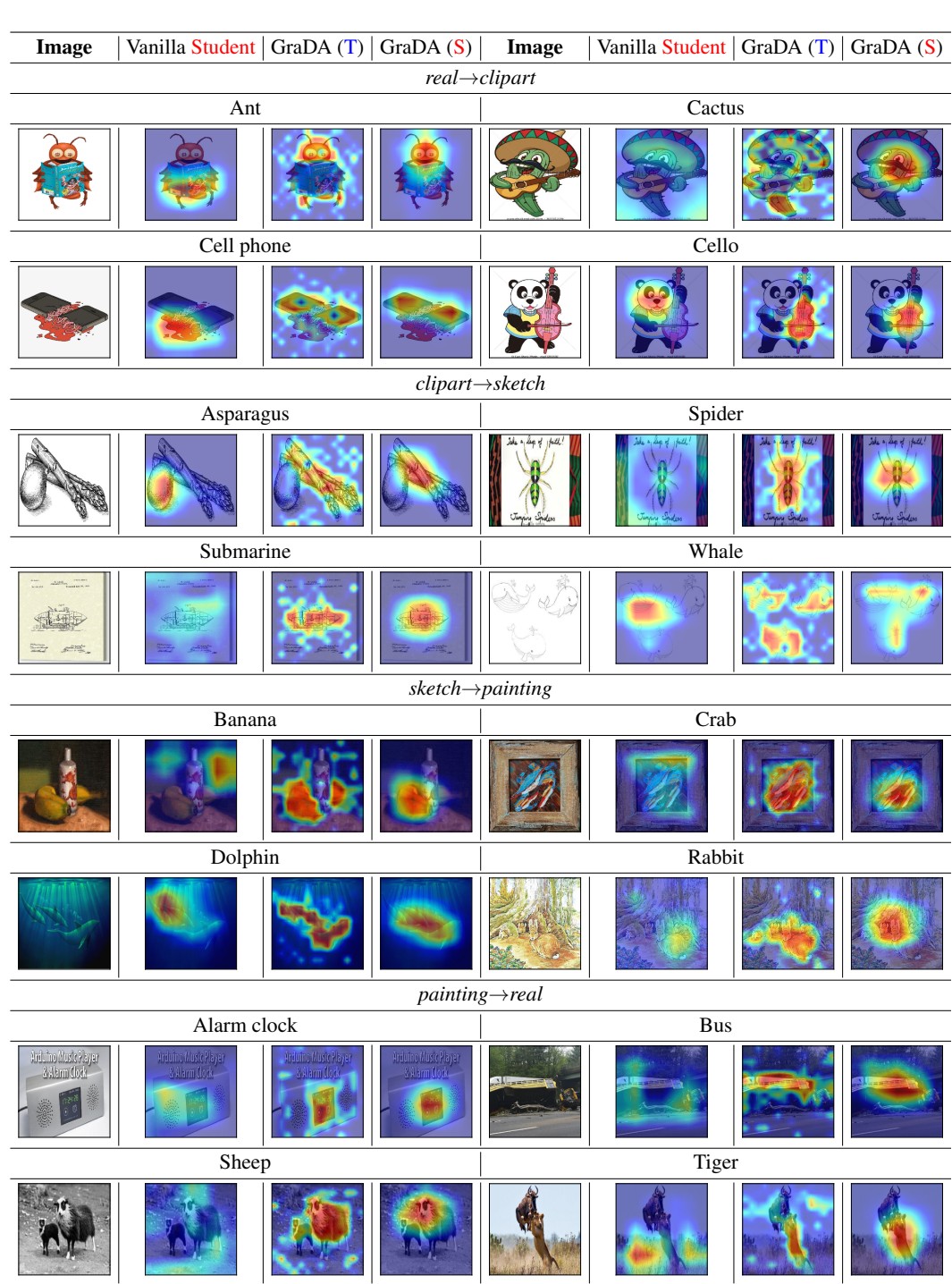

Table 20: Attention maps of the teacher and student networks on **DomainNet** in the 3-shot SSDA setting. The visualization displays class-discriminative regions in 4 diverse samples from the rel→clp, clp→skt, skt→pnt, and pnt→rel tasks.

