# OpenReview forum: "GraDA: Gradient-Guided Knowledge Distillation for Domain Adaptation"
_ICLR.cc/2026/Conference — ICLR 2026 Conference Withdrawn Submission_

### Official Review · Reviewer_68Lo · 2025-10-30

**Soundness:** 2
**Presentation:** 3
**Contribution:** 1
**Rating:** 2
**Confidence:** 5

**Summary:**

The paper investigates how to improve knowledge distillation in a domain adaptation scenario, examining the impact of the teacher network’s architecture and classification-head design on student performance under domain shift. To this end, the authors propose replacing the standard MLP head in the teacher with a Category-level Aggregation (CA) module (inspired by GCNs) to better capture relational information among samples and classes, then distill from a ViT+CA teacher to a CNN+MLP student. Extensive experiments show gains under domain shift.

**Strengths:**

1. The paper is well-written and structured, making the methods and results easy to follow.
2. The experimental evaluation is comprehensive, covering multiple baselines, ablations, and domain-shift scenarios.

**Weaknesses:**

1. The authors claim that "We identify two key factors impacting student performance under domain shift: (1) the capability of the teacher network and (2) the effectiveness of the knowledge distillation strategy." However, the statements do indeed seem trivial and already well-established in the literature, and as such, they offer very little in terms of novelty or framing a clear research question. The idea that a teacher model’s capacity (stronger network, more parameters/training) influences knowledge distillation is well-known. Similarly, the “effectiveness of the KD strategy” is also a standard concern [1].
2. The authors argue that the standard multilayer perceptron (MLP) classification head “may have limited generalization due to its inability to capture relational information among neighboring samples”, and therefore propose a Category-level Aggregation (CA) module inspired by graph convolutional networks (GCNs). However, this motivation is not sufficiently justified in the context of the proposed research question. Specifically, it is not clearly demonstrated why an MLP head would fail in the particular setting of teacher-student knowledge distillation, domain adaptation, and architecture mismatch (e.g., ViT teacher → CNN student). If the goal is to study knowledge distillation under domain shift, one could reasonably adopt an MLP head and still examine the core research question (teacher capability vs KD strategy). In other words, even with an MLP head, one could fairly compare methods and isolate the proposed components.
3. While replacing an MLP head with a Category-level Aggregation (CA) module inspired by GCNs is interesting, the novelty appears limited when viewed in the context of related work [2] [3].


[1] A survey on knowledge distillation: Recent advancements

[2] 2019 - CVPR - GCAN: Graph Convolutional Adversarial Network for Unsupervised Domain Adaptation

[3] 2020 - ECCV - Learning to Combine: Knowledge Aggregation for Multi-Source Domain Adaptation

**Questions:**

See Weakness.

---

### Official Review · Reviewer_Aed7 · 2025-10-30

**Soundness:** 3
**Presentation:** 3
**Contribution:** 2
**Rating:** 4
**Confidence:** 4

**Summary:**

This paper proposes a new framework to improve student network performance in knowledge distillation (KD) for domain adaptation (DA). The authors identify two major factors affecting KD effectiveness under domain shift: the teacher network’s capability and the distillation strategy. To address the first factor, they design a ViT+CA teacher model, which combines a ViT for rich feature extraction with a Category-level Aggregation module that uses graph-based message passing to enhance intra-class relations and reduce domain discrepancies. For the second factor, they employ pseudo labels generated by the teacher to guide the CNN+MLP student model, aligning its learning behaviour with the teacher while maintaining computational efficiency. Experiments across multiple domain adaptation benchmarks show that the proposed method achieves superior performance compared to state-of-the-art approaches.

**Strengths:**

This paper is well-recognized and easy to follow.

The core idea of enhancing the teacher module and improving knowledge distillation is intuitive and reasonable.

Experiments show consistent improvement of the proposed method applied to existing methods.

**Weaknesses:**

Why not directly use a pre-trained model with fine-tuning strategies to obtain a stronger teacher model? If this approach was intentionally avoided, please clarify the advantages of your proposed teacher model compared to pre-trained alternatives. It would strengthen the paper to include an experimental comparison with a pre-trained-based domain adaptation (DA) baseline.

The proposed class-level aggregation techniques resemble prototype-based methods, which have been extensively explored in domain adaptation. Although these techniques are introduced here in the context of knowledge distillation-based DA, please clarify the key differences and novel aspects compared to existing prototype-based approaches.

Since class-level aggregation is used to enhance intra-class relationships within unlabeled target data and generate pseudo-labels, its effectiveness likely depends on the training batch size. Have you conducted any experiments analyzing the impact of batch size on performance? Similarly, as the confidence threshold controls pseudo-label quality, please provide a hyperparameter sensitivity analysis for this threshold.

The method constructs a cross-domain knowledge graph to align unlabeled target samples with labeled source samples through class-aware feature alignment, where pseudo-labels and source ground-truth labels share identical categories. This design allows the teacher network to capture structural representations and reduce inter-domain discrepancies. Is there any theoretical justification or analysis supporting the effectiveness of this alignment strategy?

**Questions:**

Please refer to the weakness section.

---

### Official Review · Reviewer_wKQr · 2025-10-30

**Soundness:** 3
**Presentation:** 3
**Contribution:** 3
**Rating:** 6
**Confidence:** 3

**Summary:**

The paper introduces GraDA (Gradient-Guided Knowledge Distillation for Domain Adaptation), a method to improve unsupervised domain adaptation (UDA) by distilling knowledge from a powerful ViT-based teacher network to a lightweight CNN-based student network. The teacher combines a Vision Transformer (ViT) for feature extraction with a novel Category-level Aggregation (CA) module, inspired by graph convolutional networks, to enhance intra-class relations and align features across domains using pseudo labels. The student, consisting of a CNN feature extractor and MLP classifier, is trained via gradient guidance from the teacher's pseudo labels, allowing flexible learning without strict imitation.

**Strengths:**

Innovative Architecture Integration: Effectively leverages ViT's global representation strengths for training while deploying a compact CNN for inference, addressing real-world deployment challenges on resource-constrained devices.

Category-level Aggregation (CA) Module: The GCN-inspired module promotes intra-class consistency and class-aware cross-domain alignment, potentially reducing domain shift more robustly than standard MLP classifiers.

Gradient-Guided KD Strategy: Unlike traditional logit- or feature-based distillation, this method uses pseudo labels to guide gradients across all student parameters, bridging cross-architecture gaps (ViT to CNN) and allowing the student autonomy in learning, inspired by educational principles.

**Weaknesses:**

ependency on Pseudo Labels: The method heavily relies on teacher-generated pseudo labels for both self-enhancement and student guidance, which could propagate errors if initial labels are inaccurate or if the confidence threshold (τ) is poorly tuned, especially in severe domain shifts.

Computational Overhead: While the student is efficient, the ViT+CA teacher requires substantial resources during training, limiting scalability for very large datasets or low-resource environments.

**Questions:**

How sensitive is the performance to the pseudo-label confidence threshold τ? Could you provide ablation results on varying τ values across different datasets?

In the self-enhanced learning step, how does the method mitigate the impact of noisy pseudo labels on the combined dataset D_cb, especially early in training when the teacher is less reliable?

---

### Official Review · Reviewer_GLAa · 2025-10-31

**Soundness:** 3
**Presentation:** 3
**Contribution:** 2
**Rating:** 6
**Confidence:** 3

**Summary:**

This paper proposes GRADA, a gradient-guided knowledge distillation framework designed to improve student network performance under domain adaptation settings. The authors identify two major factors that limit student effectiveness during domain shift: the strength of the teacher model and the quality of the distillation process. To address these, GRADA employs a ViT-based teacher network enhanced with a Category-level Aggregation module, which strengthens intra-class relationships and reduces domain discrepancies. Knowledge is then transferred to a lighter CNN+MLP student model using pseudo labels that guide parameter updates through gradient alignment. Experiments across multiple domain adaptation benchmarks show that GRADA consistently outperforms state-of-the-art methods, while maintaining efficient inference and reduced computational cost.

**Strengths:**

The novelty of GRADA lies in its gradient-guided knowledge distillation framework that explicitly aligns student model updates with a strong ViT-based teacher enhanced by a Category-level Aggregation module. This combination uniquely strengthens intra-class relationships and reduces domain shift, enabling efficient and accurate domain adaptation with lower computational cost.

The paper demonstrates:

•	Innovative distillation approach that effectively aligns teacher and student gradients for improved domain adaptation.

•	Strong empirical performance, outperforming state-of-the-art methods across multiple benchmarks.

•	Efficiency-focused design, achieving high accuracy while reducing computational cost and inference time.

**Weaknesses:**

Some areas that could be further investigated:

•	Increased model complexity due to the ViT-based teacher and Category-level Aggregation module.

•	Limited generalization evidence beyond the evaluated domain adaptation benchmarks.

•	Clarity is impacted due to terminology being used before it is introduced. Please check the paper carefully to ensure readability. E.g. Fig 1 caption, UDA, and in Fig 1, Ours-S, Ours-T. While UDA is in the Table 6, it is common practice to write acronyms in full in the body of the paper the first time they are used. Additionally Ours-S, Ours-T is replaced by other terminology later in the paper.

**Questions:**

Can you discuss the increased model complexity due to the ViT-based teacher and Category-level Aggregation module?

Can you discuss how GRADA could be evaluated on real world datasets as opposed to benchmark datasets? Do you envisage any challenges in its use in the real world?

In the related works, you highlight the shortcomings of the current approaches. Can you describe how GRADA addresses the second shortcoming around knowledge distillation?

Personally I struggle with claims such as “Notably, the success is fully explainable …”. How do you know that it is “fully” explainable? It is more accurate to simply claim “explained by thorough qualitative analyses.” Please discuss.

Labels of Fig 3(a) and Fig 3(b) in Fig 3 do not align with their use in the text.

---

### Note · Authors · 2025-12-26

I have read and agree with the venue's withdrawal policy on behalf of myself and my co-authors.